# Extracting Common Components from Partially Observed Views Using Diffusion Geometry

**Bar Weiss**  *barweiss@campus.technion.ac.il*
*Viterbi Faculty of Electrical and Computer Engineering*
*Technion, Haifa, Israel*

**Hau-Tieng Wu**  *hw3653@nyu.edu*
*Courant Institute of Mathematical Sciences*
*New York University, New York, USA*

**Ronen Talmon**  *ronen@ef.technion.ac.il*
*Viterbi Faculty of Electrical and Computer Engineering*
*Technion, Haifa, Israel*

**Reviewed on OpenReview:** *https://openreview.net/forum?id=yTHGIV8ToF*

## Abstract

Data acquired from multiple sensors or modalities, commonly referred to as multiview data, is prevalent in real-world applications. A core problem in multiview data analysis is finding representations of common components across views while filtering out view-specific nuisance factors. A widely spread assumption in existing methods is that the views are fully aligned, where each sample has measurements from all views. However, in practice, data is often partially aligned, where some samples have missing measurements from one or more views, and only a subset of the samples are fully aligned. In this work, we propose ADM+, a multiview manifold learning algorithm that computes a low-dimensional embedding of common information from partially aligned data. ADM+ extends Alternating Diffusion Maps (ADM), an existing multiview manifold learning method, to the partial alignment setting by using fully aligned samples as anchor points for extracting common components for unaligned samples. Unlike existing methods, ADM+ does not require prior imputation of missing data or interpolation in the embedding space and makes use of all available data. We provide a computationally efficient implementation, improving upon the $O(N^3)$ time complexity of ADM, and a theoretical analysis showing that ADM+ approximates an anisotropic diffusion process that emphasizes common components. Empirical evaluations across three domains – dynamical systems, synthetic multiview images, and real-world functional magnetic resonance imaging (fMRI) – demonstrate that ADM+ achieves favorable performance compared to kernel- and manifold-based baselines. In addition, ADM+ shows robustness to distributional discrepancies between aligned and unaligned samples.

## 1 Introduction

Multiview data refers to observations captured from multiple perspectives, modalities, or sensors and is prevalent across various domains. In image processing, images from different angles enhance scene understanding (Yu et al., 2023). In autonomous systems and robotics, multi-sensor setups integrate cameras, LiDAR, and radar data for improved perception (Alatise & Hancke, 2020). Similarly, in biomedical applications, wearable devices integrate multiple multimodal sensors for health monitoring (King et al., 2017). As multiview data becomes increasingly prevalent, so does the demand for multiview data fusion methods (Lahat et al., 2015; Zhang et al., 2024), aiming to learn informative representations for each view while accounting for mutual relationships between views. Data from different views often vary significantly in the feature spaces

and dimensions, statistical properties, and similarity metrics, making fusion challenging. For example, in autonomous car navigation, cameras capture texture and color, while LiDAR provides depth measurements, each requiring different processing techniques. Moreover, different views may introduce different noise or view-specific information, which might be nuisance. For instance, when tracking an object using multiple cameras, each camera captures background objects unrelated to the target that should be excluded from the representation. A well-established approach to address nuisance factors is to extract representations of common components shared across views (Lahat et al., 2015). Unlike general multiview fusion, common-component extraction aims to isolate the shared latent structure while suppressing view-specific variability, i.e., learning an embedding invariant to view-specific factors. However, most common component extraction methods assume fully aligned views; that is, each sample has measurements from all views, with direct correspondences across them. In practice, this assumption rarely holds, as real-world data is often incomplete, with some views missing for certain samples (Wen et al., 2022). As a result, views are only partially aligned, where alignment occurs only when measurements from all views are available. For example, consider a system with two cameras capturing the same scene from different angles. Due to occlusions or sensor failures, some time instances are observed by both cameras, while others are captured by only one, so alignment is available only for the subset of jointly observed samples. This introduces challenges in determining how to effectively account for missing measurements. We refer to such data as *partial multiview data*, though the term incomplete multiview data is also commonly used in the literature. Our focus is on common-component extraction from partial multiview data using kernel and manifold learning methods.

Several kernel methods have been developed for extracting common components with partial multiview data (Wen et al., 2022). Recall that when the dataset is fully viewed, the classical approach, termed Canonical Correlation Analysis (CCA) (Hotelling, 1936), is to maximize feature correlations between views. To extend CCA to nonlinear settings, kernel methods, such as Kernel CCA (KCCA) (Fukumizu et al., 2007), map each view into a high-dimensional space, enabling the extraction of nonlinear relationships. To handle partial alignment, Trivedi et al. (2010) proposed to compute kernel entries from available measurements, impute missing entries, and then apply KCCA to the imputed kernels. However, this imputation relies on a smoothness assumption between views, which may fail when views are noisy or significantly different. Additionally, kernel imputation requires estimating many missing elements, typically leading to poor approximations. Michaeli et al. (2016) introduced Nonparametric CCA (NCCA), a kernel-based approach that extracts components with maximal mutual information. In NCCA, fully viewed samples are embedded first, and the embedding of partially observed samples is interpolated via Nyström interpolation (Williams & Seeger, 2000). However, relying solely on fully viewed samples to construct the embedding space does not make full use of the available data. Another common application of kernels is manifold learning, based on the manifold assumption that high-dimensional data often concentrate near a low-dimensional latent manifold (Belkin et al., 2006). Accordingly, measurements from each view are assumed to lie on a low-dimensional manifold embedded in a Euclidean space. In manifold learning, the kernel function captures local similarities, and the resulting kernel matrix acts as the adjacency matrix of a weighted graph approximating the geometry of the data manifold. Alternating Diffusion Maps (ADM) (Lederman & Talmon, 2018) is a manifold learning algorithm that applies a diffusion operator to the graph representation of different views, extracting common information from them. The eigenvectors and eigenvalues of the diffusion operator define an embedding of the common information. The ADM algorithm has been used in many applications (Talmon & Wu, 2019; Xiao et al., 2018; Katz et al., 2019; Román-Messina et al., 2023) and several extensions have been developed (Shnitzer et al., 2019; Yeh et al., 2025; Mendelman & Talmon, 2025). Dov et al. (2017) extended ADM to partial multiview settings using Nyström-based interpolation. However, like NCCA, it constructs the embedding from fully viewed samples and interpolates the rest, thus not fully using the data, and lacks formal justification since ADM has not been shown to admit a Nyström-compatible kernel form.

In this work, we propose a multiview manifold learning algorithm to compute a low-dimensional embedding of common information from partially aligned views. The proposed algorithm, termed ADM+, extends ADM to handle partial alignment. It leverages diffusion geometry (Coifman & Lafon, 2006) by applying a novel diffusion operator to the graph representation of the views. The diffusion operator combines both views by using the fully viewed samples as anchor points to allow information transfer between the views. While anchor points have been used in multiview fusion (Guo & Ye, 2019; Liu et al., 2024) and scalable manifold learning (Shen & Wu, 2022; Yeh et al., 2025), to our knowledge ADM+ is the first to leverage

fully aligned anchors specifically for common-component extraction under partial alignment. This design enables the use of both fully and partially viewed samples, allowing ADM+ to utilize all available data without requiring separate imputation or interpolation. The method includes a computationally efficient implementation, improving upon the $O(N^3)$ complexity of the standard ADM. We evaluate ADM+ on three tasks across diverse data domains: (i) identifying coherent sets in a dynamical system, (ii) detecting the rotation angle of a common object in synthetic multiview images, and (iii) classifying the mental activity performed by subjects during functional magnetic resonance imaging (fMRI) scans, where ADM+ is used to suppress nuisance sources of variation. Comparisons with existing kernel- and manifold-based methods show that ADM+ achieves favorable performance. In particular, using the synthetic image dataset, we demonstrate that ADM+ is more robust to discrepancies between the distributions of aligned and unaligned data, exhibiting milder performance degradation than competing methods.

We present a theoretical analysis of ADM+. Specifically, we show that, in the limit of infinite samples, ADM+ converges to an anisotropic diffusion process that emphasizes common components across views. The limiting process exhibits stronger diffusion toward aligned samples, where common-component extraction is most accurate. In addition, we examine an alternative strategy for incorporating anchors in the diffusion process, highlighting the theoretical advantages of the proposed method. Particularly, it is advantageous in handling discrepancies in the distributions of aligned and unaligned data. The structure of the paper is as follows. Sec. 2 presents related work on the use of anchors in manifold learning. Sec. 3 formulates the problem mathematically. Sec. 4 presents preliminaries on diffusion maps (DM) and ADM. In Sec. 5, we present our method, ADM+, along with complexity analysis. Sec. 6 presents the theoretical setting and analysis. In Sec. 7, we present experiments demonstrating the performance of ADM+ in three domains. Finally, Sec. 8 summarizes our conclusions.

## 2 Related Work

To contextualize our contribution, we summarize related work on using anchor points in manifold learning. The core idea is to select a subset of data points, referred to as *anchor points*, and measure the affinity between all points relative to them. This technique is primarily used in single-view settings to speed up traditional algorithms (De Silva & Tenenbaum, 2004; Belabbas & Wolfe, 2009), as it requires computing affinities only with anchor points, reducing computational and storage costs.

Shen et al. (2022) introduced a novel use of anchors, which they refer to as landmarks, to develop a fast embedding algorithm, termed RObust and Scalable Embedding via LANdmark Diffusion (ROSELAND). They defined an affinity measure through landmarks and used it to construct a diffusion operator on the data manifold, capturing the geometric structure of the data via Eigenvalue Decomposition (EVD). The landmark-based affinity produces a kernel form that enables efficient embedding computation using the Singular Value Decomposition (SVD) of a smaller rectangular matrix, accelerating the embedding process. Yeh et al. (2025) extended this idea to the multiview setting with *fully aligned* views, making the ADM algorithm faster. They introduced a low-rank diffusion operator and demonstrated that its EVD can be computed from the EVD of a small square matrix, reducing computational complexity. In the partial-view setting, Guo & Ye (2019) proposed Anchor Partial Multiview Clustering (APMC), which uses fully viewed samples as anchor points to compute a normalized affinity kernel between all points and the anchor points based on the available views. Specifically, if a sample has only one view, the affinity is computed using that view; if both views are available, the average affinity is used. This kernel is used to construct a graph, to which Laplacian Eigenmaps (Belkin & Niyogi, 2003) is applied to embed the multiview data. However, APMC targets multiview fusion rather than common-component extraction, and it mixes view-specific affinities within a single kernel, potentially obscuring the distinct geometry of each view. Finally, some partial multiview fusion methods optimize reconstruction-based objectives (Wan et al., 2024; Liu et al., 2024). These approaches are typically more computationally expensive, often requiring an SVD at each iteration; we therefore treat them as secondary baselines, but include Liu et al. (2024) for completeness. In Appendix F, we discuss related methods in more detail.

While this paper focuses on kernel and manifold learning approaches, there is also a substantial body of work on deep learning methods for partially aligned multiview data. Representative approaches include generative

models that impute missing measurements (Ma et al., 2021; Wang et al., 2023), as well as multimodal transformers that handle missing inputs via masking (Ma et al., 2022; Huan et al., 2023). Related hybrid approaches have also been proposed; for example, Yacobi et al. (2025) combine spectral embedding, CCA, and deep networks to embed partially aligned multiview data. For a comprehensive overview, we refer the reader to Wu et al. (2026). These deep learning approaches have shown strong empirical performance, but they typically require large amounts of training data across modalities and involve significant computational and memory costs, especially during training. In contrast, kernel and manifold learning methods can be more suitable in regimes with limited data or computational resources.

## 3 Problem Formulation

Consider three hidden random variables $X$, $Y$, and $Z$, with ranges supported on manifolds $\mathcal{M}_X, \mathcal{M}_Y$ and $\mathcal{M}_Z$, respectively, and a joint p.d.f $p_{X,Y,Z}(\boldsymbol{x}, \boldsymbol{y}, \boldsymbol{z})$. We model $Y$ and $Z$ as conditionally independent given $X$, where $X$ captures the information shared across the two views, and $Y$ and $Z$ represent view-specific variability. For simplicity, we assume $\mathcal{M}_X, \mathcal{M}_Y$, and $\mathcal{M}_Z$ are compact smooth manifolds without boundary. Realizations of the hidden variables are triplets $(\boldsymbol{x}_i, \boldsymbol{y}_i, \boldsymbol{z}_i)$, which we observe through two views, as follows:

$$\boldsymbol{s}_i^{(1)} = f(\boldsymbol{x}_i, \boldsymbol{y}_i), \qquad \boldsymbol{s}_i^{(2)} = g(\boldsymbol{x}_i, \boldsymbol{z}_i), \tag{1}$$

where $f$ and $g$ are unknown observation functions. In the two-camera example from Section 1, $\boldsymbol{s}_i^{(1)}$ and $\boldsymbol{s}_i^{(2)}$ correspond to images captured by the first and second cameras, respectively. The variable $X$ represents shared scene content visible in both views, while $Y$ and $Z$ capture view-specific content, such as objects visible in only one view, along with sensor noise and artifacts. We assume that the measurements lie on Riemannian manifolds embedded in Euclidean spaces, i.e., $\boldsymbol{s}_i^{(1)} \in \mathcal{M}^{(1)} \subseteq \mathbb{R}^{q_1}$ and $\boldsymbol{s}_i^{(2)} \in \mathcal{M}^{(2)} \subseteq \mathbb{R}^{q_2}$; that is, $f : \mathcal{M}_X \times \mathcal{M}_Y \to \mathcal{M}^{(1)}$ and $g : \mathcal{M}_X \times \mathcal{M}_Z \to \mathcal{M}^{(2)}$ are unknown measurable functions associated with the associated Borel sigma-algebras mapping the realizations of hidden variables, $(\boldsymbol{x}_i, \boldsymbol{y}_i, \boldsymbol{z}_i)$, into measurement pairs, $(\boldsymbol{s}_i^{(1)}, \boldsymbol{s}_i^{(2)})$, or views. Note that the first view captures realizations of only $X$ and $Y$ and the second view captures realizations of only $X$ and $Z$. Therefore, we term $X$ the *common variable* (or common information) and $Y$ and $Z$ the view-specific, or nuisance, variables. Following this model (Lederman & Talmon, 2018), each realization of the three hidden variables gives rise to a pair of *aligned* measurements. In this work, we consider the partial-view setting, where only $M$ aligned measurement pairs $\{(\boldsymbol{s}_i^{(1)}, \boldsymbol{s}_i^{(2)})\}_{i=1}^M$ are provided, along with $L$ additional measurements $\{\boldsymbol{s}_i^{(1)}\}_{i=M+1}^{M+L}$, from the first view only. We assume the $M$ aligned measurements follow a *different* p.d.f $\tilde{p}_{X,Y,Z}(\boldsymbol{x}, \boldsymbol{y}, \boldsymbol{z})$ with the same support as $p_{X,Y,Z}(\boldsymbol{x}, \boldsymbol{y}, \boldsymbol{z})$ to account for potential bias. For convenience, we denote $N = M + L$ to be the total number of measurements, including both aligned and unaligned. We refer to the $M$ aligned pairs as the full-view set, denoted by $\Omega_f$, and the $L$ additional measurements as the partial-view set, denoted by $\Omega_p$. We refer to the set of all available measurements as the unified set, denoted by $\Omega = \Omega_f \cup \Omega_p$. Our goal is to extract a low-dimensional representation $\boldsymbol{\Phi} \in \mathbb{R}^{N \times r}$ of the common variable from all measurement pairs in $\Omega$, both full and partial. Additionally, the algorithm should be computationally efficient with respect to $L$. While we present the case where only the second view is missing, Appendix A shows how to handle missing measurements in both views.

## 4 Preliminaries

In this section, we provide the necessary background for our proposed method. We begin by discussing single-view diffusion operators, showing how a diffusion process on the data manifold captures its geometric structure. We present both the continuous setting, explaining its connection to key geometric objects such as the manifold heat kernel and the Laplace–Beltrami operator. We then discuss the discrete setting with finite samples, where the process can be interpreted as a random walk on the data points and relate it to the continuous formulation. Finally, we review ADM in the fully aligned multiview setting, showing how random walks can be leveraged to extract common components across views. Our method, ADM+, presented in Section 5, extends these principles to partially aligned multiview data.

### 4.1 Single-View Diffusion Operators

Here we present preliminaries on diffusion geometry and the Diffusion Maps (DM) (Coifman & Lafon, 2006) dimensionality reduction algorithm. Let $\mathcal{M}$ be a Riemannian manifold with a metric $g$ isometrically embedded in $\mathbb{R}^q$, define the following kernel function:

$$k_\epsilon(\boldsymbol{x}, \boldsymbol{x}') = \exp\left\{-\left(d_\mathcal{M}(\boldsymbol{x}, \boldsymbol{x}')^2\right)/\epsilon\right\}, \quad \boldsymbol{x}, \boldsymbol{x}' \in \mathcal{M}, \tag{2}$$

where $d_\mathcal{M}(\cdot, \cdot)$ is the chosen metric on the manifold and $\epsilon > 0$ is the kernel scale. While other kernels can be considered, we focus on the Gaussian kernel to simplify the discussion. This choice is also motivated by the central role of the heat kernel in manifold learning. As $\epsilon$ becomes small, the Gaussian kernel approximates the manifold's heat kernel, allowing the induced graph diffusion to capture the manifold's intrinsic geometry (Coifman & Lafon, 2006). Usual choices of $d_\mathcal{M}(\cdot, \cdot)$ include the Euclidean distance or a domain-specific metric based on known properties of the data. The kernel is normalized by:

$$p_\epsilon(\boldsymbol{x}, \boldsymbol{x}') = k_\epsilon(\boldsymbol{x}, \boldsymbol{x}')/d_\epsilon(\boldsymbol{x}), \quad q_\epsilon(\boldsymbol{x}, \boldsymbol{x}') = k_\epsilon(\boldsymbol{x}, \boldsymbol{x}')/d_\epsilon(\boldsymbol{x}'), \tag{3}$$

where $d_\epsilon(\boldsymbol{x}) = \int_\mathcal{M} k_\epsilon(\boldsymbol{x}, \boldsymbol{x}')\mu(\boldsymbol{x}')dV(\boldsymbol{x}')$, $V$ is the Riemannian volume measure induced by $g$, and $\mu(\boldsymbol{x}')$ is the density function of points on $\mathcal{M}$. Nadler et al. (2006) interpret this normalization as defining transition probabilities in a Markovian process, where $q_\epsilon(\boldsymbol{x}, \boldsymbol{x}')$ represents forward-time transitions and $p_\epsilon(\boldsymbol{x}, \boldsymbol{x}')$ corresponds to backward-time transitions. Accordingly, define the backward and forward diffusion operators:

$$\mathcal{P}_\epsilon[f](\boldsymbol{x}) = \int_\mathcal{M} p_\epsilon(\boldsymbol{x}, \boldsymbol{x}')f(\boldsymbol{x}')\mu(\boldsymbol{x}')dV(\boldsymbol{x}'), \quad \mathcal{Q}_\epsilon[f](\boldsymbol{x}) = \int_\mathcal{M} q_\epsilon(\boldsymbol{x}, \boldsymbol{x}')f(\boldsymbol{x}')\mu(\boldsymbol{x}')dV(\boldsymbol{x}'), \tag{4}$$

where $f \in C^\infty(\mathcal{M})$. If $f(\boldsymbol{x})$ describes the probability density of finding a particle at location $x \in \mathcal{M}$ at time $t = 0$, then $\mathcal{Q}_\epsilon[f](\boldsymbol{x})$ is the evolution of the density to time $t = \epsilon$. Similarly, $\mathcal{P}_\epsilon[f](\boldsymbol{x})$ is analogous to conditional expectation and propagates averages of the function on the manifold. Hence, both operators diffuse functions on $\mathcal{M}$, while in different ways. Coifman & Lafon (2006) demonstrated that the spectral decomposition of $\mathcal{P}_\epsilon$ or $\mathcal{Q}_\epsilon$ allows embedding $\mathcal{M}$ into a low-dimensional Euclidean space, deriving the connection of these diffusion operators to the Laplace-Beltrami operator and the associated Fokker-Planck equation on $\mathcal{M}$. This concept forms the basis of the DM algorithm.

In the discrete case, given $N$ i.i.d measurements $\boldsymbol{x}_i \in \mathcal{M}, \quad i = 1, ..., N$, construct the kernel matrix $\boldsymbol{K} \in \mathbb{R}^{N \times N}$, where $(\boldsymbol{K})_{i,j} = k_\epsilon(\boldsymbol{x}_i, \boldsymbol{x}_j)$. Define the diagonal degree normalization matrix, analogous to $d_\epsilon$ in the continuous case, as $\boldsymbol{D} = \text{diag}(\boldsymbol{K}\boldsymbol{1}) \in \mathbb{R}^{N \times N}$, where $\boldsymbol{1} \in \mathbb{R}^N$ is a vector with all entries 1. Then normalize the kernel matrix to obtain the discretized forward and backward diffusion operators:

$$\boldsymbol{P} = \boldsymbol{D}^{-1}\boldsymbol{K}, \quad \boldsymbol{Q} = \boldsymbol{K}\boldsymbol{D}^{-1}. \tag{5}$$

These operators can be applied via matrix multiplication to any discretized function, represented as a vector $\boldsymbol{f} \in \mathbb{R}^N$. We view $\boldsymbol{f}$ as samples of a function on $\mathcal{M}$ at $\{\boldsymbol{x}_i\}_{i=1}^N$. Notably, since $\boldsymbol{K}$ is symmetric, it follows that $\boldsymbol{P} = \boldsymbol{Q}^\top$, where $[\cdot]^\top$ denotes the matrix transpose. Similar to the continuous case, the matrix $\boldsymbol{Q}$ defines Markov transition probabilities for a random walk on the data points. Accordingly, applying the operators $\boldsymbol{P}$ and $\boldsymbol{Q}$ to discrete functions retains the same forward-backward interpretation as in the continuous case.

Let $\phi_i$ and $\lambda_i$ be the left eigenvectors and eigenvalues of $\boldsymbol{Q}$, respectively, where the eigenvalues are arranged in descending order $1 = \lambda_1 \geq \lambda_2 \geq ... \geq \lambda_N \geq 0$. The eigenvalues are all non-negative since the Gaussian kernel is positive definite. The DM embedding of the data points into $\mathbb{R}^r$ is given by:

$$\boldsymbol{\Phi}_t(i) = [\lambda_2^t \phi_2(i), ..., \lambda_{r+1}^t \phi_{r+1}(i)], \quad i = 1, ..., N, \tag{6}$$

where $t$ is the diffusion time parameter denoting the number of diffusion steps applied. Since each eigenvalue correlates to the smoothness of the associated eigenvector on the data manifold, $t$ adjusts the scaling of each embedding component with respect to its smoothness. Notably, $\boldsymbol{Q}$ is similar to the symmetric matrix $\boldsymbol{D}^{-1/2}\boldsymbol{K}\boldsymbol{D}^{-1/2}$, hence its EVD is real, ensuring a real embedding. The first eigenvector is discarded as it is always proportional to $\boldsymbol{1}$. Equivalently, one may use the right eigenvectors of $\boldsymbol{P}$. A key property of the DM algorithm is that the Euclidean distance between embedded points, known as the diffusion distance, reflects

the manifold's intrinsic geometry by measuring similarity through random-walk connectivity on the data graph. This makes it robust to noise (Coifman & Lafon, 2006) and topological distortions (Bronstein et al., 2010). Diffusion distances and diffusion-map embeddings have been successfully applied in a wide range of applications (Li et al., 2017; Trstanova et al., 2020; Zhang et al., 2025).

## 4.2 Alternating Diffusion Maps

ADM extends DM to the multiview setting by defining a random walk incorporating both views. Given $N$ aligned measurements from two views $\{(\boldsymbol{s}_i^{(1)}, \boldsymbol{s}_i^{(2)})\}_{i=1}^N$, as in Sec. 3 but with full alignment. For $\ell = 1, 2$, let $\mathbf{K}_\ell \in \mathbb{R}^{N \times N}$ be a kernel matrix for each view:

$$(\mathbf{K}_\ell)_{i,j} = \exp\left\{ -\left( d_{\mathcal{M}^{(\ell)}}(\boldsymbol{s}_i^{(\ell)}, \boldsymbol{s}_j^{(\ell)})^2 \right)/\epsilon_\ell \right\}, \tag{7}$$

where $\epsilon_\ell > 0$ is the kernel scale and $d_{\mathcal{M}^{(\ell)}}(\cdot, \cdot)$ is the chosen metric. Normalizing each kernel, we get a forward diffusion operator for each view $\boldsymbol{Q}_\ell = \boldsymbol{K}_\ell \boldsymbol{D}_\ell^{-1}$,, where $\boldsymbol{D}_\ell = \mathrm{diag}(\mathbf{1}^\top \boldsymbol{K}_\ell)$. Using these operators, we define the alternating diffusion operator $\boldsymbol{Q}^{\mathrm{ADM}} = \boldsymbol{Q}_1 \boldsymbol{Q}_2$. Note that $\boldsymbol{Q}^{\mathrm{ADM}}$ is a column stochastic matrix, interpreted as transition probabilities of a random walk with alternating kernels ($\boldsymbol{Q}_2$ on odd steps and $\boldsymbol{Q}_1$ on even steps). Assuming $Y$ and $Z$ to be statistically independent given $X$, each diffusion step averages the view-specific variable related to the other view, resulting in an anisotropic diffusion process propagating along the common variable $X$. Lederman & Talmon (2018) showed that using $\boldsymbol{Q}^{\mathrm{ADM}}$, a diffusion distance metric that measures similarity based on the common variable, can be defined.

Building on this, if $\boldsymbol{Q}^{\mathrm{ADM}}$ is diagonalizable over $\mathbb{R}$, compute the embedding measurements into $\mathbb{R}^r$ using the EVD of $\boldsymbol{Q}^{ADM}$ like in equation 6. However, unlike DM, the eigenvalues and eigenvectors might not be real. In this case, taking the real part of the embedding is a common strategy for ensuring realness, at the risk of eliminating meaningful embedding information. Since $\boldsymbol{Q}^{\mathrm{ADM}}$ is not guaranteed to have a real eigenvalue decomposition. Alternatively, Talmon & Wu (2019) suggested using the SVD of $\boldsymbol{Q}^{\mathrm{ADM}}$ as follows:

$$\boldsymbol{\Phi}_t(i) = [\sigma_2^t \boldsymbol{u}_2(i), ..., \sigma_{r+1}^t \boldsymbol{u}_{r+1}(i)] \in \mathbb{R}^r, \quad i = 1, ..., N, \tag{8}$$

where $\sigma_i$ and $\boldsymbol{u}_i$ are the singular values and left singular vectors of $\boldsymbol{Q}^{\mathrm{ADM}}$ respectively, where $\sigma_i$ are decreasingly ordered. This SVD-based embedding can also capture common components in both theory and practice (Talmon & Wu, 2019). We refer to the EVD version as ADM and to the SVD version as ADM-SVD.

# 5 Proposed Method

In this section, we introduce ADM+, our proposed method for extracting common information under the partial view setting described in Sec. 3. We present the method from two perspectives: first, as a direct extension of the ADM-SVD Talmon & Wu (2019), and second, from a diffusion geometry perspective, extending the ADM diffusion process to the partial view setting.

## 5.1 Definitions and Notation

To introduce our method, we first define two affinity kernels. $\boldsymbol{K}_1^+ \in \mathbb{R}^{N \times M}$, which measures affinities between samples in $\Omega_f$ and $\Omega$ through the first view, and $\widetilde{\boldsymbol{K}_2} \in \mathbb{R}^{M \times M}$, measures affinities between samples in $\Omega_f$ in the second view.

$$(\boldsymbol{K}_1^+)_{i,m} = \exp\left\{ -\left( d_{\mathcal{M}^{(1)}}(\boldsymbol{s}_i^{(1)}, \boldsymbol{s}_m^{(1)})^2 \right)/\epsilon_1 \right\}, \quad (\widetilde{\boldsymbol{K}_2})_{k,m} = \exp\left\{ -\left( d_{\mathcal{M}^{(2)}}(\boldsymbol{s}_k^{(2)}, \boldsymbol{s}_m^{(2)})^2 \right)/\epsilon_2 \right\}, \tag{9}$$

where $i = 1, ..., N$, $k = 1, ..., M$, $m = 1, ..., M$ and $d_{\mathcal{M}^{(\ell)}}(\cdot, \cdot)$ is a chosen metric for data on $\mathcal{M}^{(\ell)}$. As in Sec. 4.1, we view matrices as operators acting on discretized functions, represented by vectors, through multiplication from the right-hand side. These functions are discretized on the data points, assigning a value to each sample in the data set. Functions on the full-view set, $\Omega_f$, are represented by vectors in $\mathbb{R}^M$ and functions on the unified set, $\Omega$, by vectors in $\mathbb{R}^N$. Hence, we use the superscript $[\cdot]^+ : \mathbb{R}^M \to \mathbb{R}^N$ to denote an operator that takes a function on $\Omega_f$ and outputs a function on $\Omega$, thus increasing the vector length.

Similarly, the superscript $[\cdot]^- : \mathbb{R}^N \to \mathbb{R}^M$ denotes a decrease in the vector length, i.e., corresponding to a transition from $\Omega$ to $\Omega_f$. Note that $(\boldsymbol{K}_1^+)^\top = \boldsymbol{K}_1^- \in \mathbb{R}^{M \times N}$. In addition, $\widetilde{[\cdot]} : \mathbb{R}^M \to \mathbb{R}^M$ denotes an operator that acts on a function defined on $\Omega_f$ and outputs a function defined on $\Omega_f$.

## 5.2 Proposed ADM+ Algorithm

The proposed ADM+ algorithm is as follows. First, we construct a rectangular matrix that captures the relationship between the full-view set $\Omega_f$ and the unified set $\Omega$. Let $\boldsymbol{Q}_1^+ \in \mathbb{R}^{N \times M}$ define a forward diffusion operator by normalizing the columns of $\mathbf{K}_1^+$:

$$\boldsymbol{Q}_1^+ = \boldsymbol{K}_1^+ (\boldsymbol{D}_1^c)^{-1}, \quad \boldsymbol{D}_1^c = \mathrm{diag}(\mathbf{1}^T \boldsymbol{D}_1^+). \quad (10)$$

The matrix $\boldsymbol{Q}_1^+$ consists of transition probabilities of a random walk from $\Omega_f$ to $\Omega$. For the second view, let $\widetilde{\boldsymbol{Q}}_2 \in \mathbb{R}^{M \times M}$ define the forward diffusion operator on the full-view set:

$$\widetilde{\boldsymbol{Q}}_2 = \widetilde{\boldsymbol{K}}_2 (\widetilde{\boldsymbol{D}}_2)^{-1}, \quad \widetilde{\boldsymbol{D}}_2 = \mathrm{diag}(\mathbf{1}^T \widetilde{\boldsymbol{K}}_2). \quad (11)$$

$\widetilde{\boldsymbol{Q}}_2$ consists of transition probabilities of a random walk on $\Omega_f$. Next, we compute $\boldsymbol{Q}_1^+ \widetilde{\boldsymbol{Q}}_2$, which is

---

**Algorithm 1** ADM+

**Require:** $\{(\boldsymbol{s}_i^{(1)}, \boldsymbol{s}_i^{(2)})\}_{i=1}^M, \{\boldsymbol{s}_i^{(1)}\}_{i=M+1}^N$, embedding dimension $r$, diffusion time $t$, kernel scales $\epsilon_\ell$

**Ensure:** $\boldsymbol{\Phi}_t \in \mathbb{R}^{N \times r}$, an embedding of the $N$ measurements into $\mathbb{R}^r$

1: Compute $\boldsymbol{K}_1^+, \widetilde{\boldsymbol{K}}_2$ using equation 9.
2: Normalize $\boldsymbol{K}_1^+, \widetilde{\boldsymbol{K}}_2$ using equation 10, equation 11 to get $\boldsymbol{Q}_1^+, \widetilde{\boldsymbol{Q}}_2$.
3: Perform SVD on $\boldsymbol{Q}_1^+ \widetilde{\boldsymbol{Q}}_2$. Obtain the left singular vectors $\boldsymbol{u}_1, \ldots, \boldsymbol{u}_{r+1}$ corresponding to the largest singular values $\sigma_1, \ldots, \sigma_{r+1}$.
4: Embed using:

$$\boldsymbol{\Phi}_t(i) = [\sigma_2^t \boldsymbol{u}_2(i), \ldots, \sigma_{r+1}^t \boldsymbol{u}_{r+1}(i)]$$

---

viewed as an anisotropic forward diffusion operator from $\Omega_f$ to $\Omega$. As stated in Sec. 3, we seek an embedding into $\mathbb{R}^r$ that represents the common variable between the views. This embedding, $\boldsymbol{\Phi}_t : \Omega \to \mathbb{R}^r$, is given using the SVD of $\boldsymbol{Q}_1^+ \widetilde{\boldsymbol{Q}}_2 \in \mathbb{R}^{N \times M}$ for the $i$-th sample by:

$$\boldsymbol{\Phi}_t(i) = [\sigma_2^t \boldsymbol{u}_2(i), ..., \sigma_{r+1}^t \boldsymbol{u}_{r+1}(i)], \quad i = 1, ..., N, \quad (12)$$

where $\sigma_i$ and $\boldsymbol{u}_i$ are the singular values and corresponding left singular vectors of $\boldsymbol{Q}_1^+ \widetilde{\boldsymbol{Q}}_2$, respectively, ordered such that $\sigma_1 \geq \sigma_2 \geq ... \geq \sigma_N$. We discard $\boldsymbol{u}_1$ and $\sigma_1$ in the embedding as is done in ADM and DM Coifman & Lafon (2006); Lederman & Talmon (2018); Talmon & Wu (2019). The parameter $t \geq 0$ controls the scaling of each dimension in the embedding, in analogy to the diffusion time parameter in ADM. In Sec. 5.3, we will present the diffusion process related to ADM+ completing this analogy. As in ADM, the singular value correlates with the smoothness of each function, $v_i$, on the common manifold $\mathcal{M}_X$. We present a pseudocode implementation of ADM+ in Algorithm 1.

Note that if all samples are aligned, $\boldsymbol{Q}_1^+ = \boldsymbol{Q}_1$ and $\widetilde{\boldsymbol{Q}}_2 = \boldsymbol{Q}_2$, and the ADM+ algorithm reduces to ADM-SVD, hence we name it ADM+. Notably, ADM+, which defines an embedding based on SVD, is equivalent to the embedding obtained by the EVD of symmetrizing $\boldsymbol{Q}_1^+ \widetilde{\boldsymbol{Q}}_2$ such that it coincides with standard spectral embedding methods. Indeed, recall that the left singular vectors of $\boldsymbol{Q}_1^+ \widetilde{\boldsymbol{Q}}_2$ are left eigenvectors of:

$$\boldsymbol{S} = \boldsymbol{Q}_1^+ \widetilde{\boldsymbol{Q}}_2 (\boldsymbol{Q}_1^+ \widetilde{\boldsymbol{Q}}_2)^T = \boldsymbol{Q}_1^+ \widetilde{\boldsymbol{Q}}_2 \widetilde{\boldsymbol{P}}_2 \boldsymbol{P}_1^-, \quad (13)$$

where $\widetilde{\boldsymbol{P}}_2 = \widetilde{\boldsymbol{Q}}_2^T$ and $\boldsymbol{P}_1^- = (\boldsymbol{Q}_1^+)^T$ are backward diffusion operators as in Sec. 4.1. Importantly, $\boldsymbol{S} \in \mathbb{R}^{N \times N}$ is a Symmetric Positive Semi-Definite (SPSD) matrix, with nonnegative eigenvalues and real eigenvectors. Similarly, ADM-SVD coincides with the EVD of $\boldsymbol{S}^{\text{ADM-SVD}} = \boldsymbol{Q}_1 \boldsymbol{Q}_2 \boldsymbol{P}_1 \boldsymbol{P}_2$. Thus, $\boldsymbol{S}$ can be viewed as a low rank extension of $\boldsymbol{S}^{\text{ADM-SVD}}$ to the partial view setting. This fact also leads to an improvement in computational complexity from $O(N^3)$ for ADM-SVD to $O(NM^2)$ for ADM+ due to computing SVD on a smaller $N \times M$ matrix. We analyze the computational complexity in detail in Appendix C.

## 5.3 Diffusion Perspective

In this section, we discuss the ADM+ algorithm from a diffusion geometry perspective. Specifically, we present the diffusion process described by the operator $\boldsymbol{S}$, equation 13. $\boldsymbol{S}$ describes an alternating diffusion process consisting of four steps, compared to the two steps in the standard ADM. These steps are illustrated

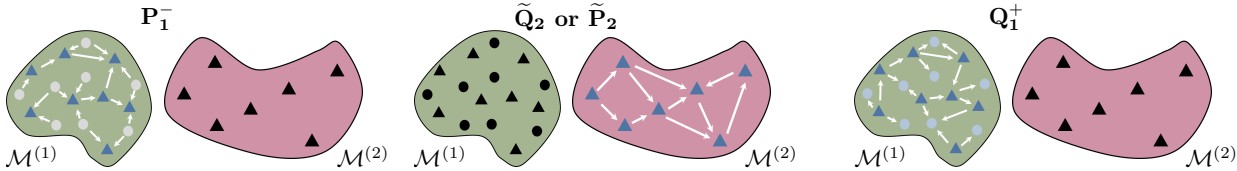

Figure 1: Illustration of the ADM+ diffusion process. Triangles and circles represent $\Omega_f$ and $\Omega_p$ samples, respectively. Samples involved in the diffusion are colored in shades of blue, with darker shades denoting higher mass concentration and gray indicating zero mass. White arrows depict mass transitions. Diffusion process steps: (left) first step $\mathbf{P}_1^-$, (center) Second and third steps $\widetilde{\mathbf{P}}_2$ and $\widetilde{\mathbf{Q}}_2$, (right) Fourth step $\mathbf{Q}_1^+$.

in Fig. 1. The first step, defined by $\boldsymbol{P}_1^-$, is a backward diffusion step from $\Omega$ to $\Omega_f$ (left subfigure). The second and third steps are backward and forward diffusion steps on $\Omega_f$, defined by $\widetilde{\boldsymbol{P}}_2$ and $\widetilde{\boldsymbol{Q}}_2$, respectively (center subfigure). The fourth and final step is a forward diffusion from $\Omega_f$ to $\Omega$, defined by $\boldsymbol{Q}_1^+$ (right subfigure). Taken together in $\boldsymbol{S}$, these four diffusion steps establish a process that diffuses between $\Omega_f$ and $\Omega$ using the first sensor affinities in $\Omega$, while incorporating the available second sensor affinities in $\Omega_f$.

The resulting diffusion process includes both forward and backward diffusion operators, contrary to the original formulation of ADM (Lederman & Talmon, 2018), which uses only forward diffusion operators. However, slightly modifying it to include only forward operators is quite straightforward. This allows to retain the original meaning of a random walk on the data rather than a general diffusion operator. As a reference, following equation 13, we define a new forward diffusion process by an $N \times N$ random walk matrix:

$$\boldsymbol{Q}_F = \boldsymbol{Q}_1^+ \widetilde{\boldsymbol{Q}}_2 \widetilde{\boldsymbol{Q}}_2 \boldsymbol{Q}_1^-, \tag{14}$$

where $\boldsymbol{Q}_1^- \in \mathbb{R}^{M \times N}$ is the forward diffusion operator from $\Omega$ to $\Omega_f$ obtained by $\boldsymbol{Q}_1^- = \boldsymbol{K}_1^- (\boldsymbol{D}_1^r)^{-1}$ $\boldsymbol{D}_1^r = \text{diag}(\mathbf{1}^T \boldsymbol{K}_1^-)$. We use the left eigenvectors and eigenvalues of $\boldsymbol{Q}_F$ to get an embedding into $\mathbb{R}^r$, as in equation 6. We term this approach as the forward-only method and provide further detail in Appendix B, including an efficient implementation and complexity analysis. Previous studies (Nadler et al., 2006; Shnitzer et al., 2019) suggest that incorporating both forward and backward diffusion operators enhances the embedding quality. Our theoretical analysis and experiments support this claim (Sec. 6 and 7).

# 6 Theoretical Analysis

Here, we present an asymptotic theoretical analysis of ADM+ under the diffusion geometry framework. Specifically, we analyze the continuous counterpart of the discrete operator $\boldsymbol{S}$ presented in Sec. 5. We show that the asymptotic expansion of $\boldsymbol{S}$ coincides with the symmetric alternating diffusion operator (Shnitzer et al., 2019) when the probability density of the full-view set matches that of the unified set. We further show that the asymptotic expansion of $\boldsymbol{S}$ suggests that ADM+ is robust to distribution discrepancies.

## 6.1 Geometric Setting

Following (Talmon & Wu, 2019; Shnitzer et al., 2019), let $\mathcal{M}^{(1)}$ and $\mathcal{M}^{(2)}$ denote the Riemannian manifolds of the two views. We assume that the manifolds are diffeomorphic via $\phi : \mathcal{M}^{(1)} \to \mathcal{M}^{(2)}$ and that they are embedded in Euclidean spaces $\mathcal{M}^{(\ell)} \subseteq \mathbb{R}^{q_\ell}$, where $\ell = 1, 2$. The diffeomorphism implies a one-to-one correspondence between the two views, ensuring that alignment is well-defined. For finite samples, this means that each sample in one view can be matched to a unique sample in the other. For each manifold, define the following kernel:

$$k_\epsilon^{(\ell)}(\boldsymbol{x}, \boldsymbol{x}') = \exp\left\{ -\left( d_{\mathcal{M}^{(\ell)}}(\boldsymbol{x}, \boldsymbol{x}')^2 \right) / \epsilon_\ell \right\}, \quad \boldsymbol{x}, \boldsymbol{x}' \in \mathcal{M}^{(\ell)} \tag{15}$$

where the distance metric $d_{\mathcal{M}^{(\ell)}}(\cdot, \cdot)$ is the Euclidean distance for the purpose of analysis. As mentioned in Sec. 3, the full-view and unified samples may be drawn from different probability distributions. Consequently, we introduce two probability densities on each manifold, $\mu^{(\ell)}$ and $\tilde{\mu}^{(\ell)}$, which model how $\Omega$ and $\Omega_f$ are

sampled, respectively. Using these densities, we define two normalization functions:

$$d_\epsilon^{(\ell)}(\boldsymbol{x}) = \int_{\mathcal{M}^{(\ell)}} k^{(\ell)}(\boldsymbol{x}, \boldsymbol{x}') \mu^{(\ell)}(\boldsymbol{x}') dV^{(\ell)}(\boldsymbol{x}'), \quad \tilde{d}_\epsilon^{(\ell)}(\boldsymbol{x}) = \int_{\mathcal{M}^{(\ell)}} k^{(\ell)}(\boldsymbol{x}, \boldsymbol{x}') \tilde{\mu}^{(\ell)}(\boldsymbol{x}') dV^{(\ell)}(\boldsymbol{x}'), \quad (16)$$

where $V^{(\ell)}$ is the Riemannian volume measure. Using these normalization functions we can normalize the kernel in four different manners, required to define the continuous counterparts:

$$p_\epsilon^{(\ell)}(\boldsymbol{x}, \boldsymbol{x}') = \frac{k_\epsilon^{(\ell)}(\boldsymbol{x}, \boldsymbol{x}')}{d_\epsilon^{(\ell)}(\boldsymbol{x})}, \quad \tilde{p}_\epsilon^{(\ell)}(\boldsymbol{x}, \boldsymbol{x}') = \frac{k_\epsilon^{(\ell)}(\boldsymbol{x}, \boldsymbol{x}')}{\tilde{d}_\epsilon^{(\ell)}(\boldsymbol{x})}, \quad q_\epsilon^{(\ell)}(\boldsymbol{x}, \boldsymbol{x}') = \frac{k_\epsilon^{(\ell)}(\boldsymbol{x}, \boldsymbol{x}')}{d_\epsilon^{(\ell)}(\boldsymbol{x}')}, \quad \tilde{q}_\epsilon^{(\ell)}(\boldsymbol{x}, \boldsymbol{x}') = \frac{k_\epsilon^{(\ell)}(\boldsymbol{x}, \boldsymbol{x}')}{\tilde{d}_\epsilon^{(\ell)}(\boldsymbol{x}')}. \quad (17)$$

## 6.2 Continuous Operator Definitions

In this section, we define the continuous counterpart of the discrete operator $\boldsymbol{S}$ from Sec. 5, following the setting described in Sec. 6.1. Since our embeddings rely on the *left* eigenvectors of these operators, and continuous operators conventionally act from the right, we analyze their transposed forms. We note that $\boldsymbol{S}^\top = \boldsymbol{S}$ due to symmetry so it remains the same. We focus here on the ADM+ algorithm but we provide similar theoretical analysis of the forward-only method in Appendix B.2.

**Definition 1.** *Let* $\mathcal{Q}_{+,\epsilon}^{(1)}, \mathcal{P}_{-,\epsilon}^{(1)} : C^\infty(\mathcal{M}^{(1)}) \to C^\infty(\mathcal{M}^{(1)})$ *be the continuous counterparts of* $\boldsymbol{Q}_1^+, \boldsymbol{P}_1^-$, *respectively, defined as:*

$$\mathcal{Q}_{+,\epsilon}^{(1)}[f](\boldsymbol{x}) = \int_{\mathcal{M}^{(1)}} q_\epsilon^{(1)}(\boldsymbol{x}, \boldsymbol{x}') f(\boldsymbol{x}') \tilde{\mu}^{(1)}(x') dV^{(1)}(\boldsymbol{x}'), \quad \mathcal{P}_{-,\epsilon}^{(1)}[f](\boldsymbol{x}) = \int_{\mathcal{M}^{(1)}} p_\epsilon^{(1)}(\boldsymbol{x}, \boldsymbol{x}') f(\boldsymbol{x}') \mu^{(1)}(\boldsymbol{x}') dV^{(1)}(\boldsymbol{x}').$$

Similarly, we define the second-view operators:

**Definition 2.** *Let* $\widetilde{\mathcal{P}}_\epsilon^{(2)}, \widetilde{\mathcal{Q}}_\epsilon^{(2)} : C^\infty(\mathcal{M}^{(2)}) \to C^\infty(\mathcal{M}^{(2)})$ *be the continuous counterparts of* $\widetilde{\boldsymbol{P}}_2, \widetilde{\boldsymbol{Q}}_2$, *respectively, defined as:*

$$\widetilde{\mathcal{P}}_\epsilon^{(2)}[f](\boldsymbol{y}) = \int_{\mathcal{M}^{(2)}} \tilde{p}_\epsilon^{(2)}(\boldsymbol{y}, \boldsymbol{y}') f(\boldsymbol{y}') \tilde{\mu}^{(2)}(\boldsymbol{y}') dV^{(2)}(\boldsymbol{y}'), \quad \widetilde{\mathcal{Q}}_\epsilon^{(2)}[f](\boldsymbol{y}) = \int_{\mathcal{M}^{(2)}} \tilde{q}_\epsilon^{(2)}(\boldsymbol{y}, vy') f(\boldsymbol{y}') \tilde{\mu}^{(2)}(\boldsymbol{y}') dV^{(2)}(\boldsymbol{y}').$$

We note that by sampling the data from $\tilde{\mu}^{(1)}$ and $\mu^{(1)}$ independently, we obtain the discrete operators. Composing the operators above with the pullback induced by the diffeomorphism $\phi$, denoted as $\phi^*$, and pushforward, $\phi_*$, define the composite operator $\mathcal{S}_\epsilon$.

**Definition 3.** *Let* $\mathcal{S}_\epsilon : C^\infty(\mathcal{M}^{(1)}) \to C^\infty(\mathcal{M}^{(1)})$ *be the continuous counterpart of* $\boldsymbol{S}$, *defined as:*

$$\mathcal{S}_\epsilon = \mathcal{Q}_{+,\epsilon}^{(1)} \phi^* \widetilde{\mathcal{Q}}_\epsilon^{(2)} \widetilde{\mathcal{P}}_\epsilon^{(2)} \phi_* \mathcal{P}_{-,\epsilon}^{(1)}. \quad (18)$$

## 6.3 Main Theorem

Below, we present the asymptotic expansion of $\mathcal{S}_\epsilon$ i.e., we express it in terms of simple differential operators.

**Theorem 1.** *When* $\epsilon_\ell$ *are sufficiently small and* $\mu^{(1)}, \tilde{\mu}^{(\ell)}$ *are sufficiently smooth, the asymptotic expansion of* $\mathcal{S}_\epsilon$ *up to order* $O(\epsilon_1^2 + \epsilon_2^2)$ *is given by:*

$$\mathcal{S}_\epsilon[f](\boldsymbol{x}) = \frac{\tilde{\mu}^{(1)}}{\mu^{(1)}} \left[ f(\boldsymbol{x}) - \epsilon_1 \left( \Delta^{(1)} f + \bar{\Delta}^{(1)} f + \frac{2\nabla^{(1)} f \cdot \nabla^{(1)} \mu^{(1)}}{\mu^{(1)}} - f \frac{\Delta^{(1)} \mu^{(1)}}{\mu^{(1)}} \right)(\boldsymbol{x}) \right.$$
$$\left. - \epsilon_2 \left( 2\phi^* \Delta^{(2)} \phi_* f + \phi^* \frac{2\nabla^{(2)} \phi_* f \cdot \nabla^{(2)} \tilde{\mu}^{(2)}}{\tilde{\mu}^{(2)}} - f \phi^* \frac{\Delta^{(2)} \tilde{\mu}^{(2)}}{\tilde{\mu}^{(2)}} \right)(\boldsymbol{x}) \right], \quad (19)$$

where $\Delta^{(\ell)}$ and $\nabla^{(\ell)}$ are the Laplace-Beltrami operator and covariant derivative of the manifold $\mathcal{M}^{(\ell)}$, respectively, and $\bar{\Delta}^{(1)} \cdot := \left( \mu^{(1)}/\tilde{\mu}^{(1)} \right) \Delta^{(1)} \left( \tilde{\mu}^{(1)}/\mu^{(1)} \right)$.

To analyze Theorem 1, we start by observing the case where $\mu^{(\ell)} = \tilde{\mu}^{(\ell)}$:

$$\mathcal{S}_\epsilon[f](\boldsymbol{x}) = f(\boldsymbol{x}) - \epsilon_1 \left( 2\Delta^{(1)} f + \frac{2\nabla^{(1)} f \cdot \nabla^{(1)} \mu^{(1)}}{\mu^{(1)}} - f \frac{\Delta^{(1)} \mu^{(1)}}{\mu^{(1)}} \right)(\boldsymbol{x})$$
$$- \epsilon_2 \left( \phi^* \Delta^{(2)} \phi_* f + \phi^* \frac{2\nabla^{(2)} \phi_* f \cdot \nabla^{(2)} \mu^{(2)}}{\mu^{(2)}} - f \phi^* \frac{\Delta^{(2)} \mu^{(2)}}{\mu^{(2)}} \right)(\boldsymbol{x}). \quad (20)$$

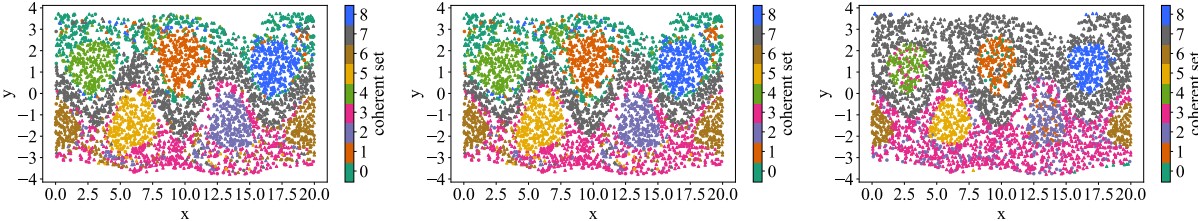

Figure 2: Identified coherent sets of the particles in Bickley jet dynamics using (left) ADM, (center) ADM+ and (right) KCCA. Colors indicate the coherent set assignment.

This expansion is equal, up to multiplication by a factor of 2 of the non-identity term, to the symmetric alternating diffusion operator ($\mathbf{S}$) presented by Shnitzer et al. (2019), which has been shown to effectively capture common components in multiview data. Suggesting that ADM+ captures common components as well when $\tilde{\mu}^{(1)} = \mu^{(1)}$. When $\tilde{\mu}^{(1)} \neq \mu^{(1)}$, note that in equation 19 there is a global factor $\tilde{\mu}^{(1)}/\mu^{(1)}$. Denoting $\hat{\mathcal{S}}_\epsilon$ such that $\mathcal{S}_\epsilon[f](\boldsymbol{x}) = (\tilde{\mu}^{(1)}/\mu^{(1)}) \cdot \hat{\mathcal{S}}_\epsilon[f](\boldsymbol{x})$ gives rise to the following asymptotic expansion:

$$
\hat{\mathcal{S}}_\epsilon[g](\boldsymbol{x}) = g(\boldsymbol{x}) - \epsilon_1 \left( \Delta^{(1)} g + \bar{\Delta}^{(1)} g + \frac{2 \nabla^{(1)} g \cdot \nabla^{(1)} \mu^{(1)}}{\mu^{(1)}} - g \frac{\Delta^{(1)} \mu^{(1)}}{\mu^{(1)}} \right)(\boldsymbol{x})
$$

$$
- \epsilon_2 \left( 2\phi^* \Delta^{(2)} \phi_* g + \phi^* \frac{2 \nabla^{(2)} \phi_* g \cdot \nabla^{(2)} \tilde{\mu}^{(2)}}{\tilde{\mu}^{(2)}} - g\phi^* \frac{\Delta^{(2)} \tilde{\mu}^{(2)}}{\tilde{\mu}^{(2)}} \right)(\boldsymbol{x}). \tag{21}
$$

This expansion of $\hat{\mathcal{S}}_\epsilon$ is similar to equation 20 with two key differences. First, $\tilde{\mu}^{(2)}$ appears instead of $\mu^{(2)}$ in the expression. This is expected as we only have access to the second view through the full-view set. Second, $\bar{\Delta}^{(1)} f$ replaces the standard Laplace-Beltrami term $\Delta^{(1)} f$. Consider the following lemma:

**Lemma 1.** *If $\psi$ is an eigenfunction of $\Delta^{(1)}$, then $\frac{\mu^{(1)}}{\tilde{\mu}^{(1)}}\psi$ is an eigenfunction of $\bar{\Delta}^{(1)}$ with the same eigenvalue.*

The proof is straightforward from the eigenfunction definition. Lemma 1 shows that $\bar{\Delta}^{(1)}$ transfers $\Delta^{(1)}$ to a different density. Following Froyland & Kwok (2020), the operator $\Delta^{(1)} + \bar{\Delta}^{(1)}$ acts as a dynamic Laplacian on the same manifold with different weighting. Thus, it extracts components that are smooth with respect to both densities $\mu^{(1)}$ and $\tilde{\mu}^{(1)}$. We illustrate this phenomenon in a simulation in Appendix D.3. Consequently, the eigenfunctions place slightly more emphasis on the unified data distribution. Due to the similarity between their asymptotic expansions with only minor differences, we can infer that $\hat{\mathcal{S}}_\epsilon$, as $\mathcal{S}_\epsilon$ captures common components, but with a slight emphasis of $\mu^{(1)}$. With this in mind, the operation of $\mathcal{S}_\epsilon$ can be understood by splitting it into two steps: (1) a diffusion step governed by $\hat{\mathcal{S}}_\epsilon$, and (2) multiplication by the weighting map $\tilde{\mu}^{(1)}/\mu^{(1)}$. In regions where $\tilde{\mu}^{(1)}(\boldsymbol{x}) \approx \mu^{(1)}(\boldsymbol{x})$ the weighting map has little effect, and the diffusion behaves as $\hat{\mathcal{S}}_\epsilon$. In regions where $\tilde{\mu}^{(1)}(\boldsymbol{x}) > \mu^{(1)}(\boldsymbol{x})$, the diffusion is amplified, while in regions where $\tilde{\mu}^{(1)}(\boldsymbol{x}) < \mu^{(1)}(\boldsymbol{x})$, it is attenuated. This mechanism drives the diffusion process toward regions with high full-view set density, where both views are available, enabling more accurate filtering of view-specific information. In Appendix D.3, we illustrate this diffusion process and compare it to the forward-only method's continuous counterpart. These visualizations contribute significantly to understanding the proposed diffusion process and are recommended for interested readers seeking further insight. This implies that ADM+ remains effective even with slight discrepancies between distributions. We provide empirical support to this claim in Sec. 7.2, with results presented in Fig. 6a. In the discrete case with finite samples, this translates to diffusion being directed toward full-view samples, enhancing the attenuation of view-specific information. Moreover, even if $\tilde{\mu}^{(1)}(\boldsymbol{x}) = \mu^{(1)}(\boldsymbol{x})$, the *finite* sample sets $\Omega_f$ and $\Omega$ may reflect different empirical distributions, further motivating the need for robustness to distributional discrepancies.

## 7 Experimental Results

In this section, we showcase the performance of ADM+ through experimental results. Our experiments involve identifying coherent sets of particles within a dynamical system, estimating the rotation angle of a common object between two images in a synthetic toy problem, and a classification task on functional MRI

(fMRI) data. Throughout all the experiments, we use the Euclidean distance to compute the kernels. The code is available on `https://github.com/barweiss1/adm_plus`.

## 7.1 Coherent Sets Identification

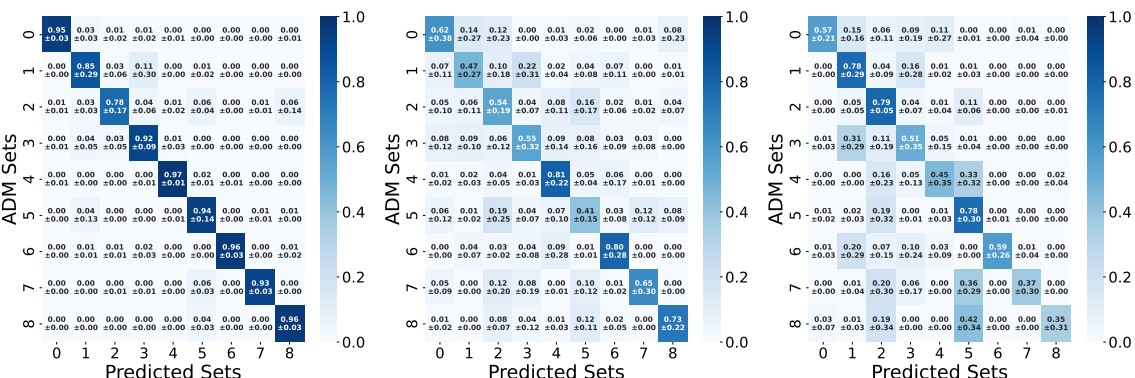

Figure 3: Confusion matrices for coherent sets of (left) ADM+ (center) NCCA (right) KCCA compared to ADM. The mean and standard deviation of each entry is reported as mean ± std.

Here, we aim to identify coherent sets of particles under fluid dynamics, specifically, the Bickley jet, a steady laminar plane jet emerging into a fluid at rest (Bickley, 1937). For details on the Bickley jet, see (Pozrikidis & Ferziger, 1997). We simulate the dynamics using the Python package Deeptime (Hoffmann et al., 2021). Our experiment follows a similar experiment reported here. We simulate a trajectory of 4000 particles in the xy-plane consisting of 401 samples in time. The flow is in the x direction, and the boundary conditions in x are periodic. Based on the trajectory, we designate two views from two snapshots of the particle positions. The first view consists of the positions at the 200-th time sample (midpoint of the trajectory). And the second view consists of the final timestamp positions. Out of the $N = 4000$ particles, we randomly select $M = 3000$ particles as our full-view set. Therefore, the first view consists of all $N = 4000$ positions, whereas the second view consists of only the subset of $M = 3000$ positions for particles in $\Omega_f$.

Given the two views, our goal is to identify the coherent sets, i.e., regions that remain coherent (non-dispersive) over time, as defined in (Froyland et al., 2010). Since particles in coherent sets remain close along their trajectories, these sets form common information between the views, even though the second view is only partially observed. Successfully handling this partiality enables assigning new particles to coherent sets using their midpoint positions alone. We apply our ADM+ to the two views, embedding each particle into 20 dimensions. In the embedded space, we use K-means to cluster the particles into 9 coherent sets. We compare ADM+ with NCCA (Michaeli et al., 2016) and KCCA (Fukumizu et al., 2007; Trivedi et al., 2010), under the same setting. To test the impact of the missing data, we also compare to ADM applied to two full views, with all $N = 4000$ measurement pairs. The kernel scale is set to $\epsilon_\ell = 1$ in all the methods, and the diffusion time to $t = 1$ in the diffusion-based methods.

Fig. 2 shows the xy coordinates of the particles at the final positions, colored according to the identified coherent sets for ADM, ADM+ and KCCA. To further illustrate the identification quality, we provide an animation of the coherent sets movement for all methods in the entire trajectory in the link. We observe that ADM and ADM+ yield similar coherent sets, indicating that ADM+ effectively mitigates the influence of the missing data. However, NCCA and KCCA show inferior identification of the coherent sets, where NCCA results in highly mixed sets, and KCCA merges two sets. The animation shows that particles assigned to the same coherent set indeed move together under the dynamic flow. To support these qualitative results, we present confusion matrices of the clustering achieved by ADM+, NCCA, and KCCA compared to the clustering achieved by ADM in Fig. 3. The results are averaged across 10 repetitions with random initial particle positions. We observe that ADM+ indeed achieves coherent sets that are the most similar to ADM.

To provide quantitative evaluation beyond direct comparison to ADM, we use two metrics: the Silhouette score (Rousseeuw, 1987), computed in the embedding space to assess cluster separation, and the dynamic

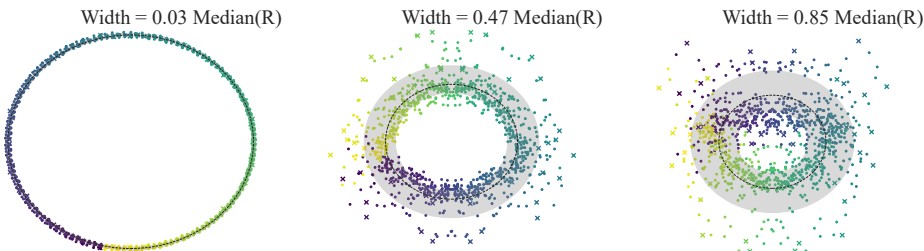

Figure 4: Embedding of the rotating images for (a) ADM (oracle, full views), (b) ADM+, and (c) Dov et al. (2017). Points are colored by the ground-truth rotation angle of the common character (down-scaled by 2.5). X marks denote $\Omega_f$ samples and circles $\Omega_p$. A dashed circle shows the median radius, and the gray ring spans the 0.15–0.85 radius quantiles; its width (in units of median radius) is reported above each subfigure.

isoperimetric ratio (Froyland, 2015; Froyland & Kwok, 2020), which measures temporal coherence:

$$I_{\mathrm{dyn}}(C) = \frac{1}{n} \sum_{i=0}^{n-1} \frac{P_{t(i)}(C)}{\min\{V_{t(i)}(C), V_{t(i)}(C^c)\}} \tag{22}$$

where $V_t(C)$ and $P_t(C)$ denote the volume and perimeter of the set $C$ at time $t$, computed from the graph affinities $K_1$ and $K_2$. We average over coherent sets and evaluate at the 200th, 250th, 300th, and final time steps; lower values indicate better coherence. We repeat the experiment 10 times and report mean $\pm$ std. The Silhouette scores for ADM, NCCA, KCCA, and ADM+ are $0.380 \pm 0.034$, $0.172 \pm 0.011$, $0.220 \pm 0.036$, and $0.409 \pm 0.016$, respectively. The corresponding dynamic isoperimetry scores are $0.375 \pm 0.017$, $0.466 \pm 0.026$, $0.493 \pm 0.047$, and $0.383 \pm 0.013$. ADM+ achieves the highest Silhouette score and the lowest dynamic isoperimetry (excluding ADM), indicating better cluster separation than NCCA and KCCA.

### 7.2 Common Objects in Images

This experiment follows the toy problem in (Lederman & Talmon, 2018). Consider a scene with three characters rotating at different rates (see Fig. 5). Two views are available, each capturing only two out of the three characters , where the snowman is captured by both views.

The objective is to estimate the rotation angle of the common character (the snowman). Importantly, we flatten each image to a long vector and do not use any image processing techniques. We generate $N = 1000$ image pairs by rotating each character at a constant, unique rate and sampling snapshots at a fixed time interval. To create the full-view set, we sample a subset of $M = 50$ image pairs, maintaining a fixed time interval between them. Thus, the first view has $N = 1000$ available images, and the second view only $M = 50$

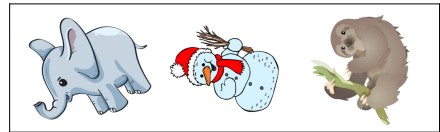

Figure 5: Rotating images setting (no downscale). Images credit to pixabay.

images. To examine the impact of common information dominance on the embedding, we manipulate the size of the common character. This is analogous to changing the signal to noise ratio (SNR) in signal processing.

Each view can be described by two periodic variables (angles), thus the manifold underlying each view is isomorphic to a 2-Torus. After filtering view-specific rotations, the common component is expected to lie on a circle. We embed the image pairs into two dimensions using (i) ADM+, (ii) the forward-only method, (iii) Dov et al. (2017), (iv) NCCA, and (v) KCCA; we additionally include (vi) ADM, with full views available, as an oracle. For diffusion-based methods, we disregard the scaling of each component by its eigenvalue, i.e., only the eigenvectors are used for embedding. which is natural for a circular embedding where components share a common scale. The kernel scale $\epsilon_\ell$ is selected by estimating the snowman's rotation angle as the polar angle in the embedding, computing the mean absolute error (MAE) against the ground truth, and choosing the scale minimize the MAE on a 20% validation subset (additional details in Appendix E.1).

Fig. 4 shows the embedding obtained by ADM, Dov et al. (2017), and ADM+ for common character down-scaled by 2.5. Appendix E.1 shows additional methods and image sizes. The points are colored by the

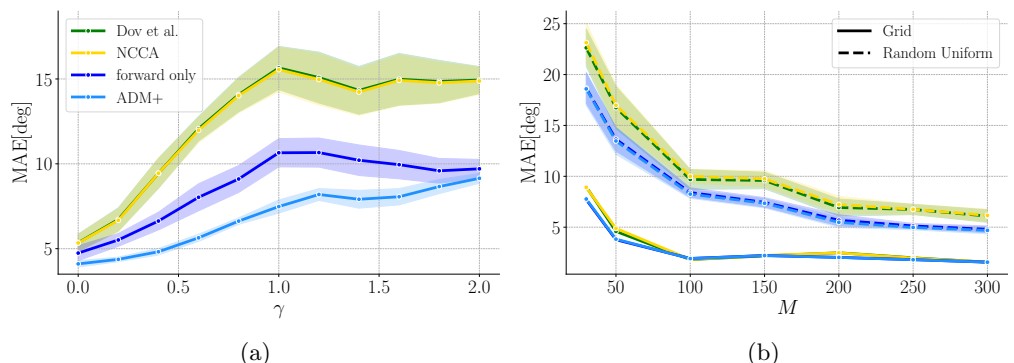

(a)                 (b)

Figure 6: (a) Angle estimation MAE vs. $\gamma$ and (b) MAE vs. $M$ for grid and random uniform sampling. For both figures the common image is down-scaled by 1.5. We present the mean across 10 repetitions with standard error (SE) confidence intervals, except for grid sampling as it is deterministic.

the rotation angle of the common character. Ideally, the embedding forms a thin circle; deviations from circularity indicate residual view-specific information. As a visual aid for the resemblance to a circle, we draw a dashed circle with the median radius of the embedded points and a shaded gray ring around it, depicting the 0.15 and 0.85 quantile radii. ADM and ADM+ capture the rotation cycle well, as can be seen by the points' colors, which change smoothly on the circle. Moreover, ADM yields a thin circle as expected, given that significantly more aligned data is available, helping suppress view-specific information. The ADM+ embedding forms a thicker ring as expected, yet it is thinner and more circular than Dov et al. (2017).

We next test robustness to distribution mismatch between $\Omega_f$ and $\Omega$ following the theoretical results (Sec. 6.3). Using the same 1000 image pairs. we sample $\Omega_f$ with $p(i) \propto (\theta_i/360)^\gamma$ with $\gamma \geq 0$, where $\theta_i$ is the rotation angle of the common character in the $i$-th pair. Thus $\Omega_f$ is biased, containing more pairs with high $\theta_i$ as $\gamma$ grows. We set $M = 100$ and, for each $\gamma$, generate 10 realizations and compute the rotation-angle MAE (down-scale factor 1.5). Fig. 6a shows the MAE, mean and STD across repetitions, for ADM+ and the competing methods. ADM+ attains the lowest error and degrades least as $\gamma$ increases, supporting the distributional robustness claim. We further examine the effect of different missingness patterns and the number of aligned measurements $M$ on the MAE. Specifically, we compare deterministic grid sampling, as used in Fig. 4, with random uniform sampling ($\gamma = 0$) over 10 repetitions. The results in Fig. 6b show that ADM+ and the forward-only method consistently outperform interpolation-based approaches NCCA and Dov et al. (2017). As expected, the grid sampling result in lower MAE as the manifold is covered better with fewer samples, this gap decreases as $M$ increases since random sampling covers the manifold better.

## 7.3 Task Classification in fMRI

We now test ADM+ on high-dimensional real fMRI data, using a predictive task. The two views are Functional Connectivity Networks (FCNs), graph representations whose nodes are brain regions and whose edge weights encode functional relationships; see Appendix E.2.1 for details. We examine the publicly available Human Connectome Project (HCP) 900 Subject Release dataset (Van Essen et al., 2013), and estimate the FCNs using the method of Gao et al. (2021). Following Gao et al. (2021), we excluded subjects with large motion artifacts, without all tasks available, or without fluid intelligence labels, remaining with fMRI scans of 506 subjects performing 9

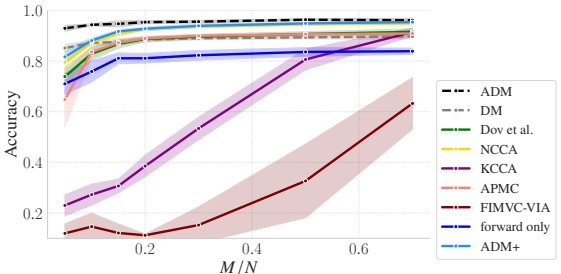

Figure 7: fMRI task classification test accuracy of different methods and $M/N$.

tasks (2 resting, 7 active). We merge the two resting tasks to a single class for a total of 8 classes. Each subject–task pair has LR and RL scans; we report LR results, with RL behaving similarly. We use all $506 \cdot 9 = 4,554$ scans and divide the data randomly into train, test, and validation sets. The validation set is always 10% of the data, the train set is set to different sizes (5-70%) and the rest is the test set. We repeat the data split 10 times. Our approach recasts the task as multiview: view 1 is the FCN of each scan, and view 2

is the FCN of a different subject performing the same task. The task label is common, while subject identity is view-specific, so common-component methods (e.g., ADM+) should suppress subject-related variability. Since constructing view 2 requires the label, it is available only for training scans, which form $\Omega_f$, making extension of the embedding to the test scans $\Omega_p$.

We compare (i) ADM+, with the following baselines. Kernel-based common-component methods: (ii) our forward-only variant, (iii) NCCA, (iv) KCCA, and (v) Dov et al. (2017); multiview fusion methods: (vi) APMC (Guo & Ye, 2019) and (vii) FIMVC-VIA (Liu et al., 2024); single-view and oracle references: (viii) DM on the first view and (ix) ADM, which uses fully aligned pairs. A 1-NN classifier is trained in the embedding space to predict the labels. For ADM, all 4,554 pairs are aligned, i.e., the test labels are hidden only during classifier training. Details on hyperparameter tuning and sensitivity are provided in Appendix E.2. The obtained classification accuracy is shown in Fig. 7. In Appendix E.2 we analyze the silhouette score, include t-SNE (Van der Maaten & Hinton, 2008) visualizations of the ADM and ADM+ embedding spaces, and compare the runtime of different methods. The results show that ADM outperforms DM, indicating that using two views is beneficial. When the second view is sufficiently represented (training size $\geq 10\%$), fusing views with ADM+ or NCCA improves accuracy, whereas KCCA, forward-only, APMC, and Dov et al. (2017) often underperform single-view DM. In particular, the multiview fusion methods APMC and FIMVC-VIA perform poorly, underscoring the importance of common-component extraction for this task. Among partial-view methods, ADM+ consistently achieves the highest accuracy across all training set sizes. Although NCCA achieves comparable accuracy in this experiment, ADM+ performs better in the previous experiments. In particular, as is shown in Fig. 6b and 7, we observe that the gap between ADM+ and NCCA increases as $M$ decreases, which is consistent with the fact that interpolation-based extensions make less efficient use of the available data.

## 8 Conclusions

In this work, we introduce ADM+, a multiview manifold learning method for finding low-dimensional representations of common components from partially aligned multiview data. ADM+ extends ADM by using fully aligned samples as anchor points within a diffusion process defined over the data manifolds. We proposed a computationally efficient implementation with $O(NM^2)$ complexity, improving upon ADM's $O(N^3)$. We present a theoretical analysis of the diffusion operator used in ADM+. Specifically, we show that the operator's continuous counterpart describes an anisotropic diffusion process that emphasizes common components and diffuses more toward fully aligned regions. This behavior increases the attenuation of view-specific information as showed in Fig. 9 and 6a. Our empirical evaluation across three domains: dynamical systems, synthetic image data, and functional MRI, demonstrates that ADM+ consistently achieves favorable performance compared to existing kernel- and manifold-based baselines. Overall, our results highlight the benefit of combining diffusion geometry with an anchor-based approach in handling partial multiview data.

Several directions remain for future work. One open question is understanding the method's dependency on $M$. While we provide empirical results in Fig. 7, it is not yet theoretically supported. Shen & Wu (2022) studied the effect of the number of landmarks (anchors) in the single-view setting, but generalizing this analysis to missing data remains an open challenge. Additionally, the role of common-information strength under partial observation remains underexplored. While Appendix E.1 presents empirical results for this case, the corresponding theory is still missing. Related analyses exist for standard ADM with fully aligned views (Ding & Wu, 2024; Katz et al., 2025), but not yet for the partial-view setting. Another interesting direction is going beyond common components. As demonstrated, different views can be used to filter out nuisance information, but they may also contain complementary information, particularly when the views originate from different modalities. Shnitzer et al. (2019) proposed a diffusion geometry-based method for extracting view-specific components in the fully aligned setting. Extending such approaches to the partial-view setting poses a significant challenge and could enhance the learned representations.

### Acknowledgments

The work of Bar Weiss and Ronen Talmon was supported by the European Union's Horizon 2020 research and innovation programme under grant agreement No. 802735-ERC-DIFFOP.

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

## A  Missing Measurements in Both Views

Our setup describes one full views and one partial throughout the paper, however the approach can be extended to include two partial views quite simply. Here we describe this extension and present some results on the image experiment data.

### A.1  Algorithm

To handle missing measurements in both views, the basic idea is to split the data into two and perform ADM+ twice. Like in the main paper we denote the number of aligned samples by $M$ but now we have additional $L_1$ samples from the first view and $L_2$ additional samples from the second view. We build two $M \times M$ kernels $\widetilde{\boldsymbol{Q}_1}$ and $\widetilde{\boldsymbol{Q}_2}$ and two rectangular kernels now with different sizes: $\boldsymbol{Q}_1^+ \in \mathbb{R}^{M+L_1 \times M}$ and $\boldsymbol{Q}_2^+ \in \mathbb{R}^{M+L_2 \times M}$. Using these kernels we can get two sets of embeddings using ADM+. $\boldsymbol{\Phi}^{(1)} \in \mathbb{R}^{M+L_1 \times r}$ from the SVD of $\boldsymbol{Q}_1^+ \widetilde{\boldsymbol{Q}_2}$ and $\boldsymbol{\Phi}^{(2)} \in \mathbb{R}^{M+L_2 \times r}$ from the SVD of $\boldsymbol{Q}_2^+ \widetilde{\boldsymbol{Q}_1}$. Now we just need to combine these two embeddings. Note that these embeddings are two sets of partially aligned multiview data, but they share the *same* geometry as they both lie on the common manifold $\mathcal{M}_X$. Thus they can be combined using APMC (Guo & Ye, 2019) or similar approaches. For each embedding define affinity matrices $\boldsymbol{K}_{\Phi 1}^+$ and $\boldsymbol{K}_{\Phi 2}^+$ with an appropriate kernel scale $\epsilon_f$, and normalize them to get $\boldsymbol{Q}_{\Phi 1}^+$ and $\boldsymbol{Q}_{\Phi 2}^+$ like in ADM+. From them, we build a rectangular matrix $\boldsymbol{A}_\Phi \in \mathbb{R}^{M+L_1+L_2 \times M}$ where the entries are:

$$
(\boldsymbol{A}_\Phi)_{i,j} = \begin{cases} \frac{1}{2}\left[(\boldsymbol{Q}_{\Phi 1}^+)_{i,j} + (\boldsymbol{Q}_{\Phi 2}^+)_{i,j}\right], & \text{if } i = 1, ..., M \\ (\boldsymbol{Q}_{\Phi 1}^+)_{i,j}, & \text{if } i = M+1, ..., M+L_1 \\ (\boldsymbol{Q}_{\Phi 2}^+)_{i-L_1,j}, & \text{if } i = M+L_1+1, ..., M+L_1+L_2 \end{cases} , i = 1, ..., N, j = 1, ..., M
$$

(23)

Combining the affinities here is justified because of the shared geometry. The affinity matrix $\boldsymbol{A}_\Phi$ contains affinity between the full view points $\Omega_f$ and the rest of the $L_1 + L_2$ partially viewed points. Therefore it can be used for algorithm like ROSELAND (Shen et al., 2022) or APMC (Guo & Ye, 2019). In APMC for example we define $\boldsymbol{D}_\Phi = \text{diag}(\boldsymbol{A}_\Phi^\top \boldsymbol{1})$, and the embedding is given by the SVD of $\boldsymbol{A}_\Phi \boldsymbol{D}_\Phi^{-1/2}$.

### A.2  Image Experiment Results

We repeat the experiment from Sec. 7.2 with the rotating characters here with missing measurements from both views. We samples $M = 50$ aligned image pairs with uniformly spaced angles from $N = 2000$ total image pairs. We equally divide the rest of the samples between the other sensor such that $L_1 = L_2 = 1025$. We apply the two missing view ADM+ algorithm with APMC for different downscale factors of the common image (1.5, 2, and 2.5). The embedding spaces are presented in Figure 8. We can see that the embedding have the shape of a ring with the angle on the ring corresponding to the rotation angle of the common image like in the single missing view experiment (Sec. 7.2). As the common image becomes smaller, the thickness of the ring increases indicating that the view-specific variables are filtered to a lesser degree.

## B  Forward-only Method

Here, we describe the forward-only method, including the algorithm description, efficient implementation and theoretical analysis.

### B.1  Implementation

Here, we describe implementation details and complexity analysis for the forward-only method. Note that $\boldsymbol{Q}_F \in \mathbb{R}^{N \times N}$ therefore performing EVD on $\boldsymbol{Q}_F$ has $O(N^3)$ time complexity. However, this complexity can be improved since $\boldsymbol{Q}_F$ has a low rank. Remind the following algebraic property:

**Remark 1.** *Given 2 matrices $\boldsymbol{M}_1 \in \mathbb{R}^{N \times M}, \boldsymbol{M}_2 \in \mathbb{R}^{M \times N}$. If $\boldsymbol{\psi}_i$ is a left eigenvector of $\boldsymbol{M}_2\boldsymbol{M}_1 \in \mathbb{R}^{M \times M}$ with eigenvalue $\lambda_i$. Then $\boldsymbol{\phi}_i = \boldsymbol{\psi}_i \boldsymbol{M}_2$ is a left eigenvector of $\boldsymbol{M}_1\boldsymbol{M}_2 \in \mathbb{R}^{N \times N}$ with the same eigenvalue $\lambda_i$.*

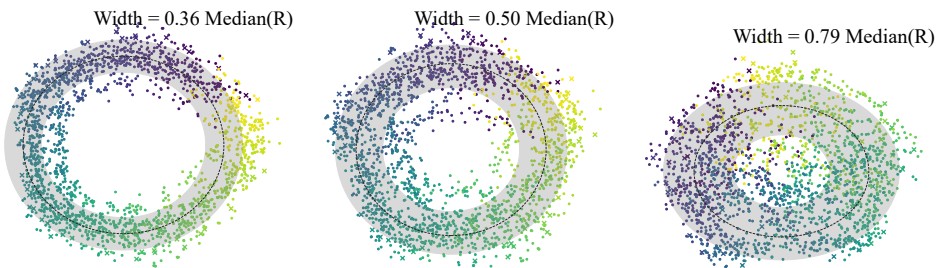

Figure 8: Embedding of the rotating images with both views missing for downscale factors: (left) 1.5, (center) 2, and (right) 2.5. Points are colored by the ground-truth rotation angle of the common character. X marks denote $\Omega_f$ samples and circles $\Omega_p$. A dashed circle shows the median radius, and the gray ring spans the 0.15–0.85 radius quantiles; its width (in units of median radius) is reported above each subfigure.

Yeh et al. (2025) used this property for a different kernel to speed up ADM. Note that for $\boldsymbol{M}_1 = \boldsymbol{Q}_1^+ \widetilde{\boldsymbol{Q}}_2$ and $\boldsymbol{M}_2 = \widetilde{\boldsymbol{Q}}_2 \boldsymbol{Q}_1^-$, we get $\boldsymbol{Q}_F = \boldsymbol{M}_1 \boldsymbol{M}_2$. Using Remark 1, we define $\hat{\boldsymbol{Q}}_F = \boldsymbol{M}_2 \boldsymbol{M}_1 = \widetilde{\boldsymbol{Q}}_2 \boldsymbol{Q}_1^- \boldsymbol{Q}_1^+ \widetilde{\boldsymbol{Q}}_2$ and we can use its EVD to compute the embedding faster. Hence, we can use this property to compute the embedding efficiently. Algorithm 2, provides pseudo-code for this efficient implementation. Appendix C analyzes the computational complexity of this implementation, showing it achieves $O(NM^2)$ complexity.

---

**Algorithm 2** Forward-only Algorithm

---

**Require:** $\{(\boldsymbol{s}_i^{(1)}, \boldsymbol{s}_i^{(2)})\}_{i=1}^M, \{\boldsymbol{s}_i^{(1)}\}_{i=M+1}^N$, embedding dimension $r$, diffusion time $t$ and kernel scales $\epsilon_1, \epsilon_2 > 0$
**Ensure:** Embedding of $N$ points into $\boldsymbol{\Phi}_t \in \mathbb{R}^{N \times l}$
1: **Compute transition matrices** $\boldsymbol{Q}_1^+, \widetilde{\boldsymbol{Q}}_2, \boldsymbol{Q}_1^-$ as described in Sec. 5.
2: **Compute** $\hat{\boldsymbol{Q}}_F = \widetilde{\boldsymbol{Q}}_2 \boldsymbol{Q}_1^- \boldsymbol{Q}_1^+ \widetilde{\boldsymbol{Q}}_2$
3: **Perform EVD on** $\hat{\boldsymbol{Q}}_F$ Get the eigenvectors $\boldsymbol{\phi}_1, ..., \boldsymbol{\phi}_{r+1}$ and eigenvalues $\lambda_1, ..., \lambda_{r+1}$, discarding the eigenvector with the largest eigenvalue $\boldsymbol{\phi}_1, \lambda_1$
4: **Get the eigenvectors of** $\boldsymbol{Q}_F$ **using Remark 1**

$$\boldsymbol{\psi}_k = \boldsymbol{\phi}_k \widetilde{\boldsymbol{Q}}_2 \boldsymbol{Q}_1^-$$

5: **Embed using**

$$\boldsymbol{\Phi}_t(i) = [\lambda_2^t \boldsymbol{\psi}_2(i), ..., \lambda_{r+1}^t \boldsymbol{\psi}_{r+1}(i)]$$

---

### B.2 Theoretical Analysis

Here, we present an asymptotic theoretical analysis of the forward-only method under the diffusion geometry framework, similarly to Sec. 6. Specifically, we analyze the continuous counterpart of the discrete operator $\boldsymbol{Q}_F$ presented in Sec. 5.3. We work under the setting defined in Sec. 6.1. We analyze transpose operators again as our embeddings rely on the *left* eigenvectors of these operators, and continuous operators conventionally act from the right. Specifically, we consider $\mathcal{P}_{F,\epsilon}$, corresponding to the forward-only method with $\boldsymbol{P}_F = \boldsymbol{Q}_F^\top = \boldsymbol{P}_1^+ \widetilde{\boldsymbol{P}}_2 \widetilde{\boldsymbol{P}}_2 \boldsymbol{P}_1^-$.

**Operator Definition**  We start by defining the continuous counterpart of $\boldsymbol{P}_F$, for which we need the following operator:

**Definition 4.** *Let $\mathcal{P}_{+,\epsilon}^{(1)} : C^\infty(\mathcal{M}^{(1)}) \to C^\infty(\mathcal{M}^{(1)})$ be the continuous counterparts of $\boldsymbol{P}_1^+$ defined as:*

$$\mathcal{P}_{+,\epsilon}^{(1)}[f](\boldsymbol{x}) = \int_{\mathcal{M}^{(1)}} \tilde{p}_\epsilon^{(1)}(\boldsymbol{x}, \boldsymbol{x}') f(\boldsymbol{x}') \tilde{\mu}^{(1)}(\boldsymbol{x}') dV^{(1)}(\boldsymbol{x}'). \tag{24}$$

Now using $\mathcal{P}_{-,\epsilon}^{(1)}$ and $\widetilde{\mathcal{P}}_{\epsilon}^{(2)}$ and the pullback $\phi_*$ and pushforward $\phi^*$ defined in Sec. 6.2. We can compose them together to define $\mathcal{P}_{F,\epsilon}$:

**Definition 5.** *Let $\mathcal{P}_{F,\epsilon} : C^\infty(\mathcal{M}^{(1)}) \to C^\infty(\mathcal{M}^{(1)})$ be the continuous counterpart of $\boldsymbol{P}_F$, defined as:*

$$\mathcal{P}_{F,\epsilon} = \mathcal{P}_{+,\epsilon}^{(1)} \phi^* \widetilde{\mathcal{P}}_{\epsilon}^{(2)} \widetilde{\mathcal{P}}_{\epsilon}^{(2)} \phi_* \mathcal{P}_{-,\epsilon}^{(1)}. \tag{25}$$

**Asymptotic Expansion**   Similarly to the analysis of $\mathcal{S}_\epsilon$ in Sec. 6.3, we analyze the asymptotic expansion of $\mathcal{P}_{F,\epsilon}$ as well. Which means expressing it in terms of simple differential operators.

**Theorem 2.** *When $\epsilon_\ell$ are sufficiently small and $\mu^{(1)}, \tilde{\mu}^{(\ell)}$ are sufficiently smooth, the asymptotic expansion of $\mathcal{P}_{F,\epsilon}$ up to order $O(\epsilon_1^2 + \epsilon_2^2)$ is given by:*

$$\mathcal{P}_{F,\epsilon}[f](\boldsymbol{x}) = f(\boldsymbol{x}) - \epsilon_1 \left( 2\Delta^{(1)} f + \frac{2\nabla^{(1)} f \cdot \nabla^{(1)} \mu^{(1)}}{\mu^{(1)}} + \frac{2\nabla^{(1)} f \cdot \nabla^{(1)} \tilde{\mu}^{(1)}}{\tilde{\mu}^{(1)}} \right)(\boldsymbol{x})$$
$$- 2\epsilon_2 \left( \phi^* \Delta^{(2)} \phi_* f + \phi^* \frac{2\nabla^{(2)} \phi_* f \cdot \nabla^{(2)} \tilde{\mu}^{(2)}}{\tilde{\mu}^{(2)}} \right)(\boldsymbol{x}). \tag{26}$$

The proof details are provided in Appendix D.2. Contrary to the expansion of $\mathcal{S}_\epsilon$, the density ratio $\tilde{\mu}^{(1)}/\mu^{(1)}$ does not appear in the expansion above. Instead, both $\mu^{(1)}$ and $\tilde{\mu}^{(1)}$ appear in the expansion terms. Therefore, the mechanism that accounts for distributional discrepancies between the anchors and the full data density does not appear here. Suggesting that the forward only method handles possible discrepancies between the densities $\mu^{(1)}$ and $\tilde{\mu}^{(1)}$ poorly. We illustrate the difference between the diffusion processes in simulation in Appendix D.3.

## C   Computational Complexity

**ADM+**   Computationally, ADM+ requires computing and storing two matrices, $\boldsymbol{Q}_1^+$ and $\widetilde{\boldsymbol{Q}}_2$, requiring $O(NM)$ space and $O(NMq_1 + M^2 q_2)$ time complexity, where $q_1$ and $q_2$ are the feature dimensions of the first and second views, respectively. Additionally, it involves a single matrix multiplication to compute $\boldsymbol{Q}_1^+ \widetilde{\boldsymbol{Q}}_2$ and applying SVD to this product. The complexity of matrix multiplication is $O(NM^2)$ corresponding to $NM$ vector multiplications of size $M$, not considering efficient implementations. The SVD of an $N \times M$ matrix also has a complexity of $O(NM^2)$. Denoting $M = N^\beta$ for $0 < \beta \leq 1$, we can express the overall time complexity neatly by $O(N^{1+2\beta} + N^{1+\beta} q_1 + N^{2\beta} q_2)$ compared to $O(N^3 + N^2 q_1 + N^2 q_2)$ of standard ADM.

**Forward Only**   The forward only method requires computing and multiplying 4 matrices to get $\hat{\mathbf{Q}}_F$ which takes $O(NMq_1 + M^2 q_2)$ for building the matrices and $O(NM^2)$ for matrix multiplications. Either for multiplying $M^2$ times vectors of length $N$ as in $\boldsymbol{Q}_1^+ \widetilde{\boldsymbol{Q}}_2$. Or for multiplying $NM$ times vectors of length $M$ as in $\boldsymbol{Q}_1^- (\boldsymbol{Q}_1^+ \widetilde{\boldsymbol{Q}}_2)$. This is without taking efficient implementations of matrix multiplications into account. We also require computing the EVD of $\hat{\mathbf{Q}}_F$ with time complexity of $O(M^3)$. Denoting $M = N^\beta$ we can express this neatly by $O(N^{1+2\beta} + N^{1+\beta} q_1 + N^{2\beta} q_2)$. We summarize the complexity in Table 1and in Appendix E.2.3 we add experimental results on runtime of different methods.

| Method | ADM | ADM+ | Forward-Only |
|---|---|---|---|
| Time Complexity | $O(N^3 + N^2 q_1 + N^2 q_2)$ | $O(NM^2 + NMq_1 + M^2 q_2)$ | $O(NM^2 + NMq_1 + M^2 q_2)$ |
| Space Complexity | $O(N^2)$ | $O(NM)$ | $O(NM)$ |

Table 1: Complexity comparison of ADM, ADM+ and forward-only.

## D   Main Theorem Proof

Here, we provide proof of the theorems presented in Sec. 6.3 and Appendix B.2.

### D.1 Single Operator Expansion

This section provides proof of the asymptotic expansion of the operators composing $\mathcal{S}_\epsilon$ and $\mathcal{P}_{F,\epsilon}$. We will compose the expansions in the following section to get the expansions of the composite operators. We cite the following Lemma from Coifman & Lafon (2006) (Appendix B, Lemma 8) to be used in our proofs in this section.

**Lemma 2.** *The asymptotic expansion of an appropriately scaled kernel $k_\epsilon$, for $x \in \mathcal{M}$, applied to any smooth function $g : \mathcal{M} \to \mathbb{R}$ is given by*

$$\int_{\mathcal{M}} k_\epsilon(\boldsymbol{x}, \boldsymbol{x}') g(\boldsymbol{x}') \, dV(\boldsymbol{x}') = m_0 g(\boldsymbol{x}) - m_2 \epsilon_1 (\Delta g(\boldsymbol{x}) - \omega(\boldsymbol{x}) g(\boldsymbol{x})) + O(\epsilon^2), \tag{27}$$

*where $m_0, m_2$ are manifold related constants, $\Delta$ is the Laplace-Beltrami operator of $\mathcal{M}$ and $\omega(\boldsymbol{x})$ is a function that depends on the curvature.*

**Lemma 3.** *The asymptotic expansion of $\mathcal{Q}_{+,\epsilon}^{(1)}$ is given by:*

$$\mathcal{Q}_{+,\epsilon}^{(1)}[f](\boldsymbol{x}) = f \frac{\tilde{\mu}^{(1)}}{\mu^{(1)}}(\boldsymbol{x}) - \frac{m_2}{m_0} \epsilon_1 \left( \Delta^{(1)} f \frac{\tilde{\mu}^{(1)}}{\mu^{(1)}} - f \frac{\tilde{\mu}^{(1)}}{\mu^{(1)}} \frac{\Delta^{(1)} \mu^{(1)}}{\mu^{(1)}} \right)(\boldsymbol{x}) + O(\epsilon_1^2), \tag{28}$$

*where $\nabla^{(1)}, \Delta^{(1)}$ are the Laplace-Beltrami operator and the covariant derivative on the manifold $\mathcal{M}^{(1)}$ respectively and $m_0, m_2$ are manifold related constants.*

*Proof.* Observe the operator definition:

$$\mathcal{Q}_{+,\epsilon}^{(1)}[f](\boldsymbol{x}) = \int_{\mathcal{M}^{(1)}} q_\epsilon^{(1)}(\boldsymbol{x}, \boldsymbol{x}') f(\boldsymbol{x}') d\tilde{\nu}^{(1)}(\boldsymbol{x}') = \int_{\mathcal{M}^{(1)}} \frac{k_\epsilon^{(1)}(\boldsymbol{x}, \boldsymbol{x}')}{d_\epsilon^{(1)}(\boldsymbol{x}')} f(\boldsymbol{x}') \tilde{\mu}^{(1)}(\boldsymbol{x}') dV^{(1)}(\boldsymbol{x}') \tag{29}$$

$$= \int_{\mathcal{M}^{(1)}} k_\epsilon^{(1)}(\boldsymbol{x}, \boldsymbol{x}') \frac{f(\boldsymbol{x}') \tilde{\mu}^{(1)}(\boldsymbol{x}')}{d_\epsilon^{(1)}(\boldsymbol{x}')} dV^{(1)}(\boldsymbol{x}') \tag{30}$$

Using Lemma 2 for $g = \frac{f \tilde{\mu}^{(1)}}{d_\epsilon^{(1)}}$, we get

$$\mathcal{Q}_{+,\epsilon}^{(1)}[f](\boldsymbol{x}) = m_0 \frac{f(\boldsymbol{x}) \tilde{\mu}^{(1)}(\boldsymbol{x})}{d_\epsilon^{(1)}(\boldsymbol{x})} - m_2 \epsilon_1 \left( \Delta^{(1)} \frac{f \tilde{\mu}^{(1)}}{d_\epsilon^{(1)}}(\boldsymbol{x}) - \omega(\boldsymbol{x}) \frac{f \tilde{\mu}^{(1)}}{d_\epsilon^{(1)}}(\boldsymbol{x}) \right) + O(\epsilon_1^2). \tag{31}$$

Now we need to find an expression for $\frac{f \tilde{\mu}^{(1)}}{d_\epsilon^{(1)}}$. Again using Lemma 2 we can expand $d_\epsilon^{(1)}(\boldsymbol{x})$

$$d_\epsilon^{(1)}(\boldsymbol{x}) = m_0 \mu^{(1)}(\boldsymbol{x}) - m_2 \epsilon_1 (\Delta^{(1)} \mu^{(1)}(\boldsymbol{x}) - \omega(\boldsymbol{x}) \mu^{(1)}(\boldsymbol{x})) + O(\epsilon_1^2). \tag{32}$$

Using the Taylor approximation of $\frac{1}{1-x}$ we get

$$\left( d_\epsilon^{(1)} \right)^{-1}(\boldsymbol{x}) = (m_0 \mu^{(1)}(\boldsymbol{x}))^{-1} \left( 1 + \frac{m_2}{m_0} \epsilon_1 \left( \frac{\Delta^{(1)} \mu^{(1)}}{\mu^{(1)}}(\boldsymbol{x}) - \omega(\boldsymbol{x}) \right) \right) + O(\epsilon_1^2). \tag{33}$$

Now we can get an expression for $\frac{f \tilde{\mu}^{(1)}}{d_\epsilon^{(1)}}$,

$$\frac{f \tilde{\mu}^{(1)}}{d_\epsilon^{(1)}}(\boldsymbol{x}) = \frac{f \tilde{\mu}^{(1)}}{m_0 \mu^{(1)}} \left( 1 + \frac{m_2}{m_0} \epsilon_1 \left( \frac{\Delta^{(1)} \mu^{(1)}}{\mu^{(1)}}(\boldsymbol{x}) - \omega(\boldsymbol{x}) \right) \right) + O(\epsilon_1^2). \tag{34}$$

Applying the Laplace-Beltrami operator we get:

$$\Delta^{(1)} \frac{f \tilde{\mu}^{(1)}}{d_\epsilon^{(1)}}(\boldsymbol{x}) = \frac{1}{m_0} \Delta^{(1)} f \frac{\tilde{\mu}^{(1)}}{\mu^{(1)}}(\boldsymbol{x}) + \frac{m_2}{m_0^2} \epsilon_1 \Delta^{(1)} \left( f \frac{\tilde{\mu}^{(1)}}{\mu^{(1)}} \left( \frac{\Delta^{(1)} \mu^{(1)}}{\mu^{(1)}}(\boldsymbol{x}) - \omega(\boldsymbol{x}) \right) \right) + O(\epsilon_1^2) \tag{35}$$

Substituting equation 34 and equation 35 into equation 31, and neglecting terms with order $O(\epsilon_1^2)$, we get:

$$\mathcal{Q}_{+,\epsilon}^{(1)}[f](\boldsymbol{x}) = f\frac{\tilde{\mu}^{(1)}}{\mu^{(1)}}(\boldsymbol{x}) - \frac{m_2}{m_0}\epsilon_1 \left(-f\frac{\tilde{\mu}^{(1)}}{\mu^{(1)}}\frac{\Delta^{(1)}\mu^{(1)}}{\mu^{(1)}} + f\frac{\tilde{\mu}^{(1)}}{\mu^{(1)}}\omega + \Delta^{(1)}f\frac{\tilde{\mu}^{(1)}}{\mu^{(1)}} - \omega f\frac{\tilde{\mu}^{(1)}}{\mu^{(1)}}\right)(\boldsymbol{x}) + O(\epsilon_1^2) \quad (36)$$

$$= f\frac{\tilde{\mu}^{(1)}}{\mu^{(1)}}(\boldsymbol{x}) - \frac{m_2}{m_0}\epsilon_1 \left(\Delta^{(1)}f\frac{\tilde{\mu}^{(1)}}{\mu^{(1)}} - f\frac{\tilde{\mu}^{(1)}}{\mu^{(1)}}\frac{\Delta^{(1)}\mu^{(1)}}{\mu^{(1)}}\right)(\boldsymbol{x}) + O(\epsilon_1^2) \quad (37)$$

□

**Lemma 4.** *The asymptotic expansion of $\mathcal{P}_{-,\epsilon}^{(1)}$ is given by:*

$$\mathcal{P}_{-,\epsilon}^{(1)}[f](\boldsymbol{x}) = f(\boldsymbol{x}) - \frac{m_2}{m_0}\epsilon_1 \left(\Delta^{(1)}f + \frac{2\nabla^{(1)}f \cdot \nabla^{(1)}\mu^{(1)}}{\mu^{(1)}}\right)(\boldsymbol{x}) + O(\epsilon_1^2), \quad (38)$$

*where $\nabla^{(1)}, \Delta^{(1)}$ are the Laplace-Beltrami operator and the covariant derivative on the manifold $\mathcal{M}^{(1)}$ respectively and $m_0, m_2$ are manifold related constants.*

*Proof.* Observe the operator definition:

$$\mathcal{P}_{-,\epsilon}^{(1)}[f](\boldsymbol{x}) = \int_{\mathcal{M}^{(1)}} p_\epsilon^{(1)}(\boldsymbol{x}, \boldsymbol{x}')f(\boldsymbol{x}')d\nu^{(1)}(\boldsymbol{x}') = \int_{\mathcal{M}^{(1)}} \frac{k_\epsilon^{(1)}(\boldsymbol{x}, \boldsymbol{x}')}{d_\epsilon^{(1)}(\boldsymbol{x})}f(\boldsymbol{x}')\mu^{(1)}(\boldsymbol{x}')dV^{(1)}(\boldsymbol{x}') \quad (39)$$

$$= \left(d_\epsilon^{(1)}(\boldsymbol{x})\right)^{-1}\int_{\mathcal{M}^{(1)}} k_\epsilon^{(1)}(\boldsymbol{x}, \boldsymbol{x}')f(\boldsymbol{x}')\mu^{(1)}(\boldsymbol{x}')dV^{(1)}(\boldsymbol{x}') \quad (40)$$

Note that in equation 33 we expanded $\left(d_\epsilon^{(1)}(\boldsymbol{x})\right)^{-1}$, hence, we need to expand the integral and compute the product. Using Lemma 2 for $g = f\mu^{(1)}$, we get

$$\int_{\mathcal{M}^{(1)}} k_\epsilon^{(1)}(\boldsymbol{x}, \boldsymbol{x}')f(\boldsymbol{x}')\mu^{(1)}(\boldsymbol{x}')dV^{(1)}(\boldsymbol{x}') = m_0 f\mu^{(1)}(\boldsymbol{x}) - m_2\epsilon_1 \left(\Delta^{(1)}f\mu^{(1)} - f\mu^{(1)}\omega\right)(\boldsymbol{x}) + O(\epsilon_1^2) \quad (41)$$

$$= m_0 f\mu^{(1)}(\boldsymbol{x}) - m_2\epsilon_1 \left(\mu^{(1)}\Delta^{(1)}f + 2\nabla^{(1)}f \cdot \nabla^{(1)}\mu^{(1)} + f\Delta^{(1)}\mu^{(1)} - f\mu^{(1)}\omega\right)(\boldsymbol{x}) + O(\epsilon_1^2) \quad (42)$$

$$= f\left[m_0\mu^{(1)} - m_2\epsilon_1 \left(\Delta^{(1)}\mu^{(1)} - \mu^{(1)}\omega\right)\right](\boldsymbol{x}) - m_2\epsilon_1 \left(\mu^{(1)}\Delta^{(1)}f + 2\nabla^{(1)}f \cdot \nabla^{(1)}\mu^{(1)}\right)(\boldsymbol{x}) + O(\epsilon_1^2) \quad (43)$$

$$\underset{\text{equation equation 32}}{=} f d_\epsilon^{(1)}(\boldsymbol{x}) - m_2\epsilon_1 \left(\mu^{(1)}\Delta^{(1)}f + 2\nabla^{(1)}f \cdot \nabla^{(1)}\mu^{(1)}\right)(\boldsymbol{x}) + O(\epsilon_1^2) \quad (44)$$

Multiplying this by $\left(d_\epsilon^{(1)}(\boldsymbol{x})\right)^{-1}$ we get the following:

$$\mathcal{P}_{-,\epsilon}^{(1)}[f](\boldsymbol{x}) = f(\boldsymbol{x}) - \left(d_\epsilon^{(1)}(\boldsymbol{x})\right)^{-1} m_2\epsilon_1 \left(\mu^{(1)}\Delta^{(1)}f + 2\nabla^{(1)}f \cdot \nabla^{(1)}\mu^{(1)}\right)(\boldsymbol{x}) + O(\epsilon_1^2) \quad (45)$$

$$\underset{\text{equation equation 33}}{=} f(\boldsymbol{x}) - \frac{m_2}{m_0}\epsilon_1 \left(\Delta^{(1)}f + \frac{2\nabla^{(1)}f \cdot \nabla^{(1)}\mu^{(1)}}{\mu^{(1)}}\right)(\boldsymbol{x}) + O(\epsilon_1^2) \quad (46)$$

□

For the forward-only method, we need to expand $\mathcal{P}_{+,\epsilon}^{(1)}$ as well.

**Lemma 5.** *The asymptotic expansion of $\mathcal{P}_{+,\epsilon}^{(1)}$ is given by:*

$$\mathcal{P}_{+,\epsilon}^{(1)}[f](\boldsymbol{x}) = f(\boldsymbol{x}) - \frac{m_2}{m_0}\epsilon_1 \left(\Delta^{(1)}f + \frac{2\nabla^{(1)}f \cdot \nabla^{(1)}\tilde{\mu}^{(1)}}{\tilde{\mu}^{(1)}}\right)(\boldsymbol{x}) + O(\epsilon_1^2). \quad (47)$$

*Proof.* Observe the operator definition:

$$\mathcal{P}_{+,\epsilon}^{(1)}[f](\boldsymbol{x}) = \int_{\mathcal{M}^{(1)}} \tilde{p}_\epsilon^{(1)}(\boldsymbol{x}, \boldsymbol{x}') f(\boldsymbol{x}') d\tilde{\nu}^{(1)}(\boldsymbol{x}') = \int_{\mathcal{M}^{(1)}} \frac{k_\epsilon^{(1)}(\boldsymbol{x}, \boldsymbol{x}')}{\tilde{d}_\epsilon^{(1)}(\boldsymbol{x})} f(\boldsymbol{x}') \tilde{\mu}^{(1)}(\boldsymbol{x}') dV^{(1)}(\boldsymbol{x}') \tag{48}$$

$$= \left(\tilde{d}_\epsilon^{(1)}(\boldsymbol{x})\right)^{-1} \int_{\mathcal{M}^{(1)}} k_\epsilon^{(1)}(\boldsymbol{x}, \boldsymbol{x}') f(\boldsymbol{x}') \tilde{\mu}^{(1)}(\boldsymbol{x}') dV^{(1)}(\boldsymbol{x}') \tag{49}$$

Note that this is the same expression as in equation 40 just with the distribution $\tilde{\mu}$ instead of $\mu$, hence the expansion is given by substituting them in the result:

$$\mathcal{P}_{+,\epsilon}^{(1)}[f](\boldsymbol{x}) = f(\boldsymbol{x}) - \frac{m_2}{m_0}\epsilon_1 \left(\Delta^{(1)}f + \frac{2\nabla^{(1)}f \cdot \nabla^{(1)}\tilde{\mu}^{(1)}}{\tilde{\mu}^{(1)}}\right)(\boldsymbol{x}) + O(\epsilon_1^2) \tag{50}$$

□

For the second manifold operators, substitute the distributions and operators to fit the second view manifold full view set.

**Lemma 6.** *The asymptotic expansion of $\widetilde{\mathcal{Q}}_\epsilon^{(2)}$ is given by:*

$$\widetilde{\mathcal{Q}}_\epsilon^{(2)}(\boldsymbol{x}) = f(\boldsymbol{x}) - \frac{m_2}{m_0}\epsilon_2 \left(\Delta^{(2)}f - f\frac{\Delta^{(2)}\tilde{\mu}^{(2)}}{\tilde{\mu}^{(2)}}\right)(\boldsymbol{x}) + O(\epsilon_2^2), \tag{51}$$

*where $\nabla^{(2)}, \Delta^{(2)}$ are the Laplace-Beltrami operator and the covariant derivative on the manifold $\mathcal{M}^{(2)}$ respectively and $m_0, m_2$ are manifold related constants.*

**Lemma 7.** *The asymptotic expansion of $\widetilde{\mathcal{P}}_\epsilon^{(2)}$ is given by:*

$$\widetilde{\mathcal{P}}_\epsilon^{(2)}[f](\boldsymbol{x}) = f(\boldsymbol{x}) - \frac{m_2}{m_0}\epsilon_2 \left(\Delta^{(2)}f + \frac{2\nabla^{(2)}f \cdot \nabla^{(2)}\tilde{\mu}^{(2)}}{\tilde{\mu}^{(2)}}\right)(\boldsymbol{x}) + O(\epsilon_2^2), \tag{52}$$

*where $\nabla^{(2)}, \Delta^{(2)}$ are the Laplace-Beltrami operator and the covariant derivative on the manifold $\mathcal{M}^{(2)}$ respectively and $m_0, m_2$ are manifold related constants.*

### D.2 Composite Operator Expansion

Here we show the asymptotic expansion of the composite operators $\mathcal{S}_\epsilon$ and $\mathcal{P}_{F,\epsilon}$. Starting with the ADM+ operator. Following Shnitzer et al. (2019), from now on we assume that the kernels are scaled such that $m_0 = m_2 = 1$ for each operator for the sake of simplicity and readability. The proof technique here is to compose the operators in the definition of each composite operator step by step with their asymptotic expansions from Section D.1. For our proofs we require the following Lemma:

**Lemma 8.** *Let $\phi$ be a pushforward/pullback operator originating from a smooth isomorphic transformation $T : \mathcal{M} \to \tilde{\mathcal{M}}$. Then for any two functions $f, g \in C^\infty(\mathcal{M})$:*

$$\phi[f \cdot g] = \phi[f] \cdot \phi[g] \tag{53}$$

*Proof.* Observing the left-hand side of the lemma with the definition of the pullback operator proves this Lemma.

$$\phi[f \cdot g] = f(T^{-1}(\boldsymbol{x})) \cdot g(T^{-1}(\boldsymbol{x})) = \phi[f] \cdot \phi[g]$$

□

$\mathcal{S}_\epsilon$ **Expansion**   Here we present the proof of Theorem 1. the expansion of the operator $\mathcal{S}_\epsilon$.

*Proof.* First, we compose $\mathcal{P}^{(1)}_{-,\epsilon}$ with $\widetilde{\mathcal{P}}^{(2)}_\epsilon$ and the pushforward $\phi_*$. To specify the domain of each function explicitly, we will denote the input of functions on the first view manifold by $x \in \mathcal{M}^{(1)}$, and functions on the second view manifold with $y \in \mathcal{M}^{(2)}$ inputs. Denote $\mathcal{G}[f](\boldsymbol{y}) = \widetilde{\mathcal{P}}^{(2)}_\epsilon \phi_* \mathcal{P}^{(1)}_{-,\epsilon}[f](\boldsymbol{y})$ and $h(\boldsymbol{y}) = \phi_* \mathcal{P}^{(1)}_{-,\epsilon}[f](\boldsymbol{y})$.

$$\mathcal{G}[f](\boldsymbol{y}) = \widetilde{\mathcal{P}}^{(2)}_\epsilon[h(\boldsymbol{y})] \tag{54}$$

$$= h(\boldsymbol{y}) - \epsilon_2 \left( \Delta^{(2)} h(\boldsymbol{y}) + \frac{2\nabla^{(2)} h \cdot \nabla^{(2)} \tilde{\mu}^{(2)}}{\tilde{\mu}^{(2)}}(\boldsymbol{y}) \right) + O(\epsilon_2^2) \tag{55}$$

$$= \phi_* f(\boldsymbol{y}) - \phi_* \epsilon_1 \left( \Delta^{(1)} f + \frac{2\nabla^{(1)} f \cdot \nabla^{(1)} \mu^{(1)}}{\mu^{(1)}} \right)(\boldsymbol{y}) - \tag{56}$$

$$- \epsilon_2 \left( \Delta^{(2)} \phi_* f(\boldsymbol{y}) + \frac{2\nabla^{(2)} \phi_* f \cdot \nabla^{(2)} \tilde{\mu}^{(2)}}{\tilde{\mu}^{(2)}}(\boldsymbol{y}) \right) + O(\epsilon_1^2 + \epsilon_2^2)$$

Now we compose the result with $\widetilde{\mathcal{Q}}^{(2)}_\epsilon$ and the pullback $\phi^*$. Define $\mathcal{H}[f](\boldsymbol{x}) = \phi^* \widetilde{\mathcal{Q}}^{(2)}_\epsilon \mathcal{G}[f](\boldsymbol{x})$, define $v(\boldsymbol{y}) = \mathcal{G}[f](\boldsymbol{y})$.

$$\mathcal{H}[f](\boldsymbol{x}) = \phi^* \widetilde{\mathcal{Q}}^{(2)}_\epsilon[v](\boldsymbol{x}) \tag{57}$$

$$= \phi^* v(\boldsymbol{x}) - \phi^* \epsilon_2 \left( \Delta^{(2)} v(\boldsymbol{x}) - v \frac{\Delta^{(2)} \tilde{\mu}^{(2)}}{\tilde{\mu}^{(2)}}(\boldsymbol{x}) \right) + O(\epsilon_2^2) \tag{58}$$

$$= f(\boldsymbol{x}) - \epsilon_1 \left( \Delta^{(1)} f + \frac{2\nabla^{(1)} f \cdot \nabla^{(1)} \mu^{(1)}}{\mu^{(1)}} \right)(\boldsymbol{x}) -$$

$$- \epsilon_2 \left( \phi^* \Delta^{(2)} \phi_* f(\boldsymbol{x}) + \phi^* \frac{2\nabla^{(2)} \phi_* f \cdot \nabla^{(2)} \tilde{\mu}^{(2)}}{\tilde{\mu}^{(2)}}(\boldsymbol{x}) \right) \tag{59}$$

$$- \epsilon_2 \left( \phi^* \Delta^{(2)} \phi_* f(\boldsymbol{x}) - \phi^* \left[ \phi_* f \frac{\Delta^{(2)} \tilde{\mu}^{(2)}}{\tilde{\mu}^{(2)}} \right](\boldsymbol{x}) \right) + O(\epsilon_1^2 + \epsilon_2^2)$$

Using Lemma 8:

$$= f(\boldsymbol{x}) - \epsilon_1 \left( \Delta^{(1)} f + \frac{2\nabla^{(1)} f \cdot \nabla^{(1)} \mu^{(1)}}{\mu^{(1)}} \right)(\boldsymbol{x}) - \tag{60}$$

$$- \epsilon_2 \left( 2\phi^* \Delta^{(2)} \phi_* f(\boldsymbol{x}) + \phi^* \frac{2\nabla^{(2)} \phi_* f \cdot \nabla^{(2)} \tilde{\mu}^{(2)}}{\tilde{\mu}^{(2)}}(\boldsymbol{x}) - f\phi^* \frac{\Delta^{(2)} \tilde{\mu}^{(2)}}{\tilde{\mu}^{(2)}}(\boldsymbol{x}) \right) + O(\epsilon_1^2 + \epsilon_2^2)$$

And finally, we compose $\mathcal{H}$ with the reference to total operator. Define $u(\boldsymbol{x}) = \mathcal{H}[f](\boldsymbol{x})$.

$$\mathcal{S}_\epsilon[f](\boldsymbol{x}) = \mathcal{Q}^{(1)}_{+,\epsilon}[u](\boldsymbol{x}) \tag{61}$$

$$= u\frac{\tilde{\mu}^{(1)}}{\mu^{(1)}}(\boldsymbol{x}) - \epsilon_1 \left( \Delta^{(1)} u\frac{\tilde{\mu}^{(1)}}{\mu^{(1)}} - u\frac{\tilde{\mu}^{(1)}}{\mu^{(1)}} \frac{\Delta^{(1)} \mu^{(1)}}{\mu^{(1)}} \right)(\boldsymbol{x}) + O(\epsilon_1^2) \tag{62}$$

$$= f\frac{\tilde{\mu}^{(1)}}{\mu^{(1)}}(\boldsymbol{x}) - \epsilon_1 \left( \frac{\tilde{\mu}^{(1)}}{\mu^{(1)}} \Delta^{(1)} f + \frac{\tilde{\mu}^{(1)}}{\mu^{(1)}} \frac{2\nabla^{(1)} f \cdot \nabla^{(1)} \mu^{(1)}}{\mu^{(1)}} \right)(\boldsymbol{x}) -$$

$$- \epsilon_2 \frac{\tilde{\mu}^{(1)}}{\mu^{(1)}} \left( 2\phi^* \Delta^{(2)} \phi_* f(\boldsymbol{x}) + \phi^* \frac{2\nabla^{(2)} \phi_* f \cdot \nabla^{(2)} \tilde{\mu}^{(2)}}{\tilde{\mu}^{(2)}}(\boldsymbol{x}) - f\phi^* \frac{\Delta^{(2)} \tilde{\mu}^{(2)}}{\tilde{\mu}^{(2)}}(r) \right) - \tag{63}$$

$$- \epsilon_1 \left( \Delta^{(1)} f\frac{\tilde{\mu}^{(1)}}{\mu^{(1)}} - f\frac{\tilde{\mu}^{(1)}}{\mu^{(1)}} \frac{\Delta^{(1)} \mu^{(1)}}{\mu^{(1)}} \right)(\boldsymbol{x}) + O(\epsilon_1^2 + \epsilon_2^2)$$

$$= \frac{\tilde{\mu}^{(1)}}{\mu^{(1)}} \left[ f(\boldsymbol{x}) - \epsilon_1 \left( \Delta^{(1)} f + \frac{\mu^{(1)}}{\tilde{\mu}^{(1)}} \Delta^{(1)} \frac{\tilde{\mu}^{(1)}}{\mu^{(1)}} f + \frac{2\nabla^{(1)} f \cdot \nabla^{(1)} \mu^{(1)}}{\mu^{(1)}} - f\frac{\Delta^{(1)} \mu^{(1)}}{\mu^{(1)}} \right)(\boldsymbol{x}) - \right. \tag{64}$$

$$\left. - \epsilon_2 \left( 2\phi^* \Delta^{(2)} \phi_* f(\boldsymbol{x}) + \phi^* \frac{2\nabla^{(2)} \phi_* f \cdot \nabla^{(2)} \tilde{\mu}^{(2)}}{\tilde{\mu}^{(2)}}(\boldsymbol{x}) - f\phi^* \frac{\Delta^{(2)} \tilde{\mu}^{(2)}}{\tilde{\mu}^{(2)}}(\boldsymbol{x}) \right) \right] + O(\epsilon_1^2 + \epsilon_2^2)$$

$\square$

$\mathcal{P}_{F,\epsilon}$ **Expansion**   Here we present the proof of Theorem 2, the expansion of the operator $\mathcal{P}_{F,\epsilon}$.

*Proof.* First, we compose $\mathcal{P}^{(1)}_{-,\epsilon}$ with $\widetilde{\mathcal{P}}^{(2)}_{\epsilon}$ and the pushforward $\phi_*$. To specify the domain of each function explicitly, we will denote the input of functions on the first view manifold by $x \in \mathcal{M}^{(1)}$, and functions on the second view manifold with $y \in \mathcal{M}^{(2)}$ inputs. Denote $\mathcal{G}[f](\boldsymbol{y}) = \widetilde{\mathcal{P}}^{(2)}_{\epsilon} \phi_* \mathcal{P}^{(1)}_{-,\epsilon}[f](\boldsymbol{y})$ and $h(\boldsymbol{y}) = \phi_* \mathcal{P}^{(1)}_{-,\epsilon}[f](\boldsymbol{y})$.

$$\mathcal{G}[f](\boldsymbol{y}) = \widetilde{\mathcal{P}}^{(2)}_{\epsilon}[h(\boldsymbol{y})] \tag{65}$$

$$= h(\boldsymbol{y}) - \epsilon_2 \left( \Delta^{(2)} h(\boldsymbol{y}) + \frac{2\nabla^{(2)} h \cdot \nabla^{(2)} \tilde{\mu}^{(2)}}{\tilde{\mu}^{(2)}}(\boldsymbol{y}) \right) + O(\epsilon_2^2) \tag{66}$$

$$= \phi_* f(\boldsymbol{y}) - \phi_* \epsilon_1 \left( \Delta^{(1)} f + \frac{2\nabla^{(1)} f \cdot \nabla^{(1)} \mu^{(1)}}{\mu^{(1)}} \right)(\boldsymbol{y}) - \tag{67}$$

$$- \epsilon_2 \left( \Delta^{(2)} \phi_* f(\boldsymbol{y}) + \frac{2\nabla^{(2)} \phi^* f \cdot \nabla^{(2)} \tilde{\mu}^{(2)}}{\tilde{\mu}^{(2)}}(\boldsymbol{y}) \right) + O(\epsilon_1^2 + \epsilon_2^2)$$

Now we compose the result with $\widetilde{\mathcal{P}}^{(2)}_{\epsilon}$ and the pullback $\phi^*$. Define $\mathcal{H}[f](\boldsymbol{x}) = \phi^* \widetilde{\mathcal{P}}^{(2)}_{\epsilon} \mathcal{G}[f](\boldsymbol{x})$, define $v(\boldsymbol{y}) = \mathcal{G}[f](\boldsymbol{y})$.

$$\mathcal{H}[f](\boldsymbol{x}) = \phi^* \widetilde{\mathcal{Q}}^{(2)}_{\epsilon}[v](\boldsymbol{x}) \tag{68}$$

$$= \phi^* v(\boldsymbol{x}) - \phi^* \epsilon_2 \left( \Delta^{(2)} v + \frac{2\nabla^{(2)} v \cdot \nabla^{(2)} \tilde{\mu}^{(2)}}{\tilde{\mu}^{(2)}} \right)(\boldsymbol{x}) + O(\epsilon_2^2) \tag{69}$$

$$= f(\boldsymbol{x}) - \epsilon_1 \left( \Delta^{(1)} f + \frac{2\nabla^{(1)} f \cdot \nabla^{(1)} \mu^{(1)}}{\mu^{(1)}} \right)(\boldsymbol{x}) - \tag{70}$$

$$- \epsilon_2 \left( \phi^* \Delta^{(2)} \phi_* f(\boldsymbol{x}) + \phi^* \frac{2\nabla^{(2)} \phi_* f \cdot \nabla^{(2)} \tilde{\mu}^{(2)}}{\tilde{\mu}^{(2)}}(\boldsymbol{x}) \right)$$

$$- \epsilon_2 \left( \phi^* \Delta^{(2)} \phi_* f(\boldsymbol{x}) + \phi^* \frac{2\nabla^{(2)} \phi_* f \cdot \nabla^{(2)} \tilde{\mu}^{(2)}}{\tilde{\mu}^{(2)}}(\boldsymbol{x}) \right) + O(\epsilon_1^2 + \epsilon_2^2)$$

$$= f(\boldsymbol{x}) - \epsilon_1 \left( \Delta^{(1)} f + \frac{2\nabla^{(1)} f \cdot \nabla^{(1)} \mu^{(1)}}{\mu^{(1)}} \right)(\boldsymbol{x}) - \tag{71}$$

$$- 2\epsilon_2 \left( \phi^* \Delta^{(2)} \phi_* f(\boldsymbol{x}) + \phi^* \frac{2\nabla^{(2)} \phi_* f \cdot \nabla^{(2)} \tilde{\mu}^{(2)}}{\tilde{\mu}^{(2)}}(\boldsymbol{x}) \right) + O(\epsilon_1^2 + \epsilon_2^2)$$

And finally, we compose $\mathcal{H}$ with $\mathcal{P}^{(1)}_{+,\epsilon}$. Define $u(\boldsymbol{x}) = \mathcal{H}[f](\boldsymbol{x})$.

$$\mathcal{P}_{F,\epsilon}[f](\boldsymbol{x}) = \mathcal{P}^{(1)}_{+,\epsilon}[u](\boldsymbol{x}) = u(\boldsymbol{x}) - \epsilon_1 \left( \Delta^{(1)} u + \frac{2\nabla^{(1)} u \cdot \nabla^{(1)} \tilde{\mu}^{(1)}}{\tilde{\mu}^{(1)}} \right)(\boldsymbol{x}) + O(\epsilon_1^2) = \tag{72}$$

$$u(\boldsymbol{x}) - \epsilon_1 \left( \Delta^{(1)} u + \frac{2\nabla^{(1)} u \cdot \nabla^{(1)} \tilde{\mu}^{(1)}}{\tilde{\mu}^{(1)}} \right)(\boldsymbol{x}) + O(\epsilon_1^2) = \tag{73}$$

$$f(\boldsymbol{x}) - \epsilon_1 \left( \Delta^{(1)} f + \frac{2\nabla^{(1)} f \cdot \nabla^{(1)} \mu^{(1)}}{\mu^{(1)}} \right)(\boldsymbol{x}) - \epsilon_1 \left( \Delta^{(1)} f + \frac{2\nabla^{(1)} f \cdot \nabla^{(1)} \tilde{\mu}^{(1)}}{\tilde{\mu}^{(1)}} \right)(\boldsymbol{x}) \tag{74}$$

$$- 2\epsilon_2 \left( \phi^* \Delta^{(2)} \phi_* f(\boldsymbol{x}) + \phi^* \frac{2\nabla^{(2)} \phi_* f \cdot \nabla^{(2)} \tilde{\mu}^{(2)}}{\tilde{\mu}^{(2)}}(\boldsymbol{x}) \right) + O(\epsilon_1^2 + \epsilon_2^2) =$$

$$f(\boldsymbol{x}) - \epsilon_1 \left( 2\Delta^{(1)} f + \frac{2\nabla^{(1)} f \cdot \nabla^{(1)} \mu^{(1)}}{\mu^{(1)}} + \frac{2\nabla^{(1)} f \cdot \nabla^{(1)} \tilde{\mu}^{(1)}}{\tilde{\mu}^{(1)}} \right)(\boldsymbol{x}) \tag{75}$$

$$- 2\epsilon_2 \left( \phi^* \Delta^{(2)} \phi_* f(\boldsymbol{x}) + \phi^* \frac{2\nabla^{(2)} \phi_* f \cdot \nabla^{(2)} \tilde{\mu}^{(2)}}{\tilde{\mu}^{(2)}}(\boldsymbol{x}) \right) + O(\epsilon_1^2 + \epsilon_2^2)$$

$$\square$$

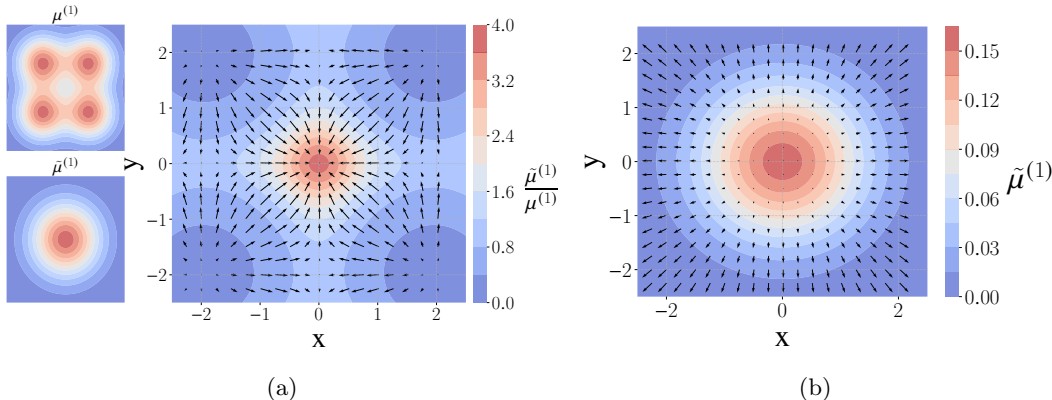

Figure 9: Single-view simulation of the expansions of (a) $\mathcal{S}_\epsilon$ and (b) $\mathcal{P}_{F,\epsilon}$. The simulation was done with $\tilde{\mu}^{(1)}(\boldsymbol{x})$ as wide Gaussian at (0, 0) and a mixture of 4 Gaussians as $\mu^{(1)}(\boldsymbol{x})$ as displayed in the left side of (a). The background of the direction plots is the distribution ratio $\frac{\tilde{\mu}^{(1)}}{\mu^{(1)}}(\boldsymbol{x})$. The arrows in (b) are enlarged by a factor of 10 to make them visible.

### D.3 Theory Illustrations

**ADM+ and Forward-only Comparison** Here we illustrate the differences between $\mathcal{S}_\epsilon$ and $\mathcal{P}_{F,\epsilon}$, through a numerical simulation of these diffusion operators in a simplified single-view setting in two-dimensional space. The simulation illustrates the weighted diffusion mechanism explained in Sec. 6.3 compared to the forward-only diffusion. Our domain is a square with periodic boundary conditions to satisfy our assumption of a manifold without boundary. We assume that the views are equal i.e., $\tilde{\mu}^{(2)} = \tilde{\mu}^{(1)}, \mu^{(2)} = \mu^{(1)}, \Delta^{(2)} = \Delta^{(1)}, \nabla^{(2)} = \nabla^{(1)}$. $\tilde{\mu}^{(1)}$ is set to a wide Gaussian distribution centered at $(0,0)$, while $\mu^{(1)}$ is set to a mixture of four Gaussians, as shown in Fig. 9a. Notably, these distributions share similar supports but have different shapes. To compute the diffusion direction, we place a delta (impulse) measure at each grid point and simulate the diffusion process for a fixed time interval using equations 19 and 26. The final location is the center of mass of the absolute value of the diffused function. The direction is then displayed as an arrow from the initial delta location to the final location, as presented in Fig. 9. The results show that $\mathcal{S}_\epsilon$ diffuses strongly towards regions where $\tilde{\mu}^{(1)}$ is high. While $\mathcal{P}_{F,\epsilon}$ diffuses down the direction of the distribution $\tilde{\mu}^{(1)}$ as in single-view DM, ignoring the different densities. The diffusion is much weaker for $\mathcal{P}_{F,\epsilon}$, as the arrows are enlarged in Fig. 9b compared to Fig. 9a by a factor of 10 for visibility. In the direction plot, black dots indicate that the arrow points to roughly the same spot, signifying isotropic diffusion, as expected for uniform Euclidean geometry. This occurs for $\mathcal{S}_\epsilon$ in regions where the density ratio remains nearly constant. For $\mathcal{P}_{F,\epsilon}$, isotropic diffusion appears near the peak of $\tilde{\mu}^{(1)}$, but in other regions the arrows are quite small (as the arrows are significantly enlarged in Fig. 9) indicating near isotropic diffusion. Thus the forward-only method nearly ignores the distribution discrepancy motivating our choice of $\mathcal{S}_\epsilon$ over $\mathcal{P}_{F,\epsilon}$.

**Eigenfunctions Illustration** Following Section 6.3, we present simulation results illustrating Lemma 1. Using the same simulation setting presented above, we show the eigenfunctions of $\Delta^{(1)}$, $\bar{\Delta}^{(1)}$ and $\frac{1}{2}(\Delta^{(1)} + \bar{\Delta}^{(1)})$, illustrating that the eigenfunctions of $\bar{\Delta}^{(1)}$ indeed highlight the density ratio $\frac{\mu^{(1)}}{\tilde{\mu}^{(1)}}$. We use a power method approach to compute the eigenfunction with the smallest eigenvalue of each operator. We start with a function $f_0(\boldsymbol{x})$ initialized as a constant, then at each iteration we apply the operator $I - \Delta^{(1)}$, where $I$ is the identity operator repeatedly and normalize between iterations by the $L_2$ norm of $f_t(\boldsymbol{x})$, as in the following equation:

$$f_{t+1}(\boldsymbol{x}) = \frac{(f_t - \Delta^{(1)} f_t)(\boldsymbol{x})}{\|(f_t - \Delta^{(1)} f_t)(\boldsymbol{x})\|_{L_2}} \tag{76}$$

And the same for $\bar{\Delta}^{(1)}$ and $\frac{1}{2}(\Delta^{(1)} + \bar{\Delta}^{(1)})$, just substituting them with $\Delta^{(1)}$ in the equation. The results of this iterative procedure for all operators are shown in Figure 10. The figure illustrates that the eigenfunctions of $\bar{\Delta}$ emphasize regions where the density $\mu^{(1)}$ is greater then the full-view set density $\tilde{\mu}^{(1)}$ as is stated by

**Lemma 1.** The eigenfunctions of the dynamic Laplacian $\frac{1}{2}(\Delta^{(1)} + \bar{\Delta}^{(1)})$ emphasize smoothness both with respect to both weightings (uniform and $\frac{\mu^{(1)}}{\bar{\mu}^{(1)}}$), thus we get a more uniform version of the eigenfunction of $\bar{\Delta}^{(1)}$.

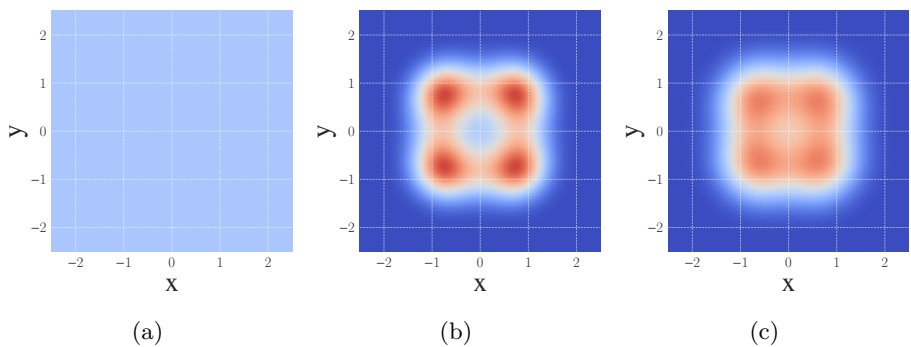

Figure 10: Absolute value of the eigenfunctions of (a) $\Delta^{(1)}$, (b) $\bar{\Delta}^{(1)}$ and (c) $\frac{1}{2}(\Delta^{(1)} + \bar{\Delta}^{(1)})$.

## E    Additional Experimental Results

In this section, we provide additional experimental results in the experiments presented in Section 7.

### E.1    Common Objects in Images

Here, we show additional results for the rotating images experiment presented in Section 7.2. Specifically, we present the embedding obtained by our forward-only method, NCCA, and KCCA in Figure 11. The results are under the same setting as Section 7.2. We can see that all these methods yield wider circles compared to ADM+. In the embedding obtain by the forward-only method, a ring is still noticeable, while in NCCA and KCCA it is distorted and the ring is not recognizable.

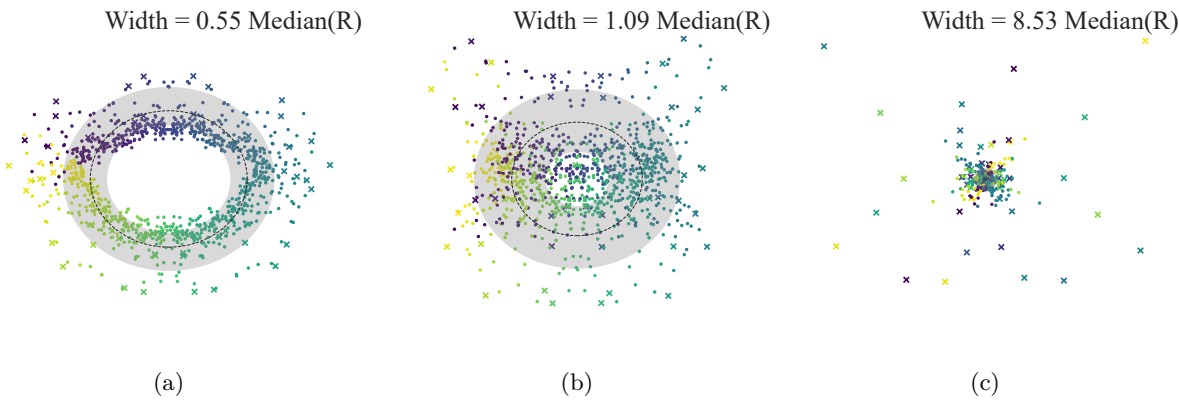

Figure 11: Embedding result for the image rotation data for the following algorithms: (a) forward-only (b) NCCA Michaeli et al. (2016), and (c) KCCA. The points are colored by the ground truth rotation angle of the common character. The common character here was down-scaled by 2.5.

To further evaluate the embedding quantitatively, we present the MAE of the rotation angle estimation obtained by the different methods. Table 2 shows the MAE of all aforementioned embedding methods for different sizes of the common character to test each algorithm's sensitivity to CMR changes. The CMR is controlled by down-scaling the common character in both axes by different down-scale (DS) factors. As expected, ADM yields the best results, as it has access to the full data, but its results provide an intuitive

performance "bound" for the other methods. Among the methods under the partial view setting, ADM+ yields the best results for all downscale factor values except DS factor=2.5, where it is a close second after our forward-only method.

**Hyperparameter Tuning** For all methods, the kernel scale is $\epsilon_\ell = a \cdot \epsilon_\ell^0$ where $\epsilon_\ell^0$ is the median of the pairwise distances in the view and $a$ is a hyperparameter (selected from $a = \{0.5, 1, 2, 3, ..., 10\}$). We select $a$ by estimating the snowman's rotation angle from the embedding (as the polar angle), computing the mean absolute error (MAE) against the ground truth while accounting for rotation and reflection symmetries, and choosing the $a$ that minimizes the MAE on a 20% validation subset.

| Method | DS factor=1.5 MAE [deg] | DS factor=2 MAE [deg] | DS factor=2.5 MAE [deg] |
|---|---|---|---|
| ADM (Full Data) | 0.86 | 1.16 | 1.28 |
| Dov et al. (2017) | 2.55 | 3.58 | 11.07 |
| NCCA (Michaeli et al., 2016) | 2.59 | 3.77 | 25.77 |
| KCCA (Trivedi et al., 2010) | 5.34 | 40.73 | 37.98 |
| forward only | 3.56 | 4.31 | **5.94** |
| ADM+ (ours) | **2.11** | **3.28** | 6.07 |

Table 2: Rotation angle estimation error for different embedding methods for rotation images data with different down-scale (DS) factors of the common image size. The lowest error value (excluding ADM) for each DS factor value is in bold.

## E.2 Task Classification in fMRI

### E.2.1 Background on FCNs

FCNs were used in several downstream tasks, such as functional connectome fingerprinting (Finn et al., 2015) and connectome-based predictive modeling (Shen et al., 2017). In (Xiao et al., 2018), information fusion from two FCNs was presented using ADM to perform IQ prediction. Here, we demonstrate the classification of mental tasks performed by subjects during fMRI scans. There are many ways to estimate FCN from fMRI signals, (Finn et al., 2015; Gao et al., 2021; Abbas et al., 2021). we use the method presented in (Gao et al., 2021), as it yields good results on the data set we examine in several downstream tasks.

### E.2.2 Hyperparameter Tuning and Sensitivity

**Hyperparameter Tuning** We tune the hyperparameters by selecting the parameters with the best accuracy on the validation set from a grid of values. Specifically, $\epsilon_\ell = a \cdot \epsilon_\ell^0$ where $\epsilon_\ell^0$ is the median of distances and $a = \{0.1, 0.5, 1, 2, 10\}$ is a hyperparameter, diffusion time $t = \{0.1, 0.2, 0.3, 0.5, 1, 2\}$, and embedding dimension $r = \{5, 10, 20, 30, ..., 100\}$.

**Hyperparameter Sensitivity** We present here results on the hyperparameter sensitivity of ADM+ compared to other methods on the fMRI data. Figure 12 presents the sensitivity to the embedding dimension $l$, kernel scale $\epsilon_\ell$, and diffusion time $t$. We choose present results for $M/N = 0.3$ because the accuracy is high for all methods at this point but still most of the data is incomplete, however other ratios show similar trends. For the embedding dimension, observe that ADM+ is much more robust than non-diffusion methods like NCCA and APMC, but the performance does drop significantly when the dimension grows too large. This happens for all methods but some methods like Dov et al. (2017), and forward-only show less degradation. We note that these methods are also more robust to the diffusion time parameter $t$, which can help in robustness to dimensionality. We see that ADM+ is very sensitive to this parameter, which possibly means that ADM+ estimates the smoothness of each eigenvector (based on singular values) worse than other methods. Additionally, it allows for the inclusion of less smooth coordinates with reduced influence on the

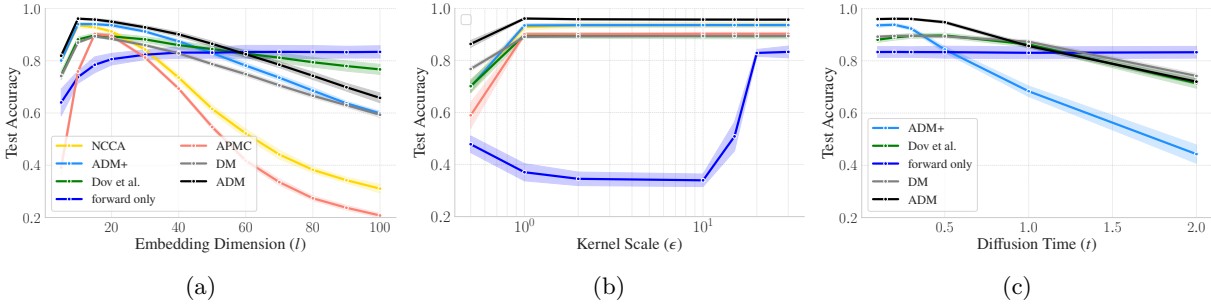

Figure 12: Hyperparameter sensitivity analysis of ADM+ compared to different methods. (a) Embedding dimension $l$, (b) kernel scale $\epsilon_\ell$ and (c) diffusion time $t$.

embedding. For the kernel scale $\epsilon_\ell$ we see that ADM+'s robustness to it is quite similar to other methods and beyond very small scales it achieves the same accuracy.

### E.2.3 Additional Results

Here, we present additional results from the experiment detailed in Section 7.3. In Figure 13 we present the Silhouette score (Rousseeuw, 1987) in the embedding space of each method for different train set sizes using the ground truth labels. The Silhouette score is presented for the hyperparameters with the best validation accuracy, as in Figure 7. We observe that, ADM, ADM+, and NCCA almost always achieve the highest silhouette score, where ADM achieves slightly higher score in correlation to the accuracy results.

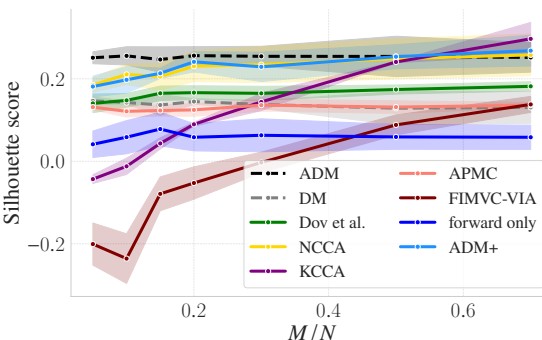

Figure 13: fMRI task Silhoeutte score of different methods. The standard deviation is used as the confidence interval.

To visually compare the embedding obtained by ADM+ and ADM, we apply t-SNE (Van der Maaten & Hinton, 2008), with perplexity of 30 and 1000 iterations, to the embedding space. Figure 14 shows the t-SNE visualizations, where the points are colored by the true labels. The embedding with ADM+ achieves a natural clustering of the different tasks. These clusters are well-separated and the partial-view samples are evenly distributed among the full-view samples. We also see that the clusters and their relative positioning are similar between ADM and ADM+, indicating that the embedding space is extended properly to partial-view points. It is important to note that t-SNE is a stochastic algorithm, so the positioning of distant clusters and the orientation of the domain may vary, as is seen between Figure 14a and 14b.

We also present the runtime results of the different methods as a function of $M/N$ in Figure 15. The experiments were conducted on a server with two AMD EPYC 7742 processors. First we observe that the FIMVC-VIA has significantly higher runtime. This is because this method works in the feature space which in this case is very high dimensional as each sample is an $259 \times 259$ matrix. Additionally, it runs for up to 30 iterations to converge which also increases the runtime significantly. Among other method, we can observe that ADM+ and forward-only achieve faster runtime than DM and ADM as expected from the

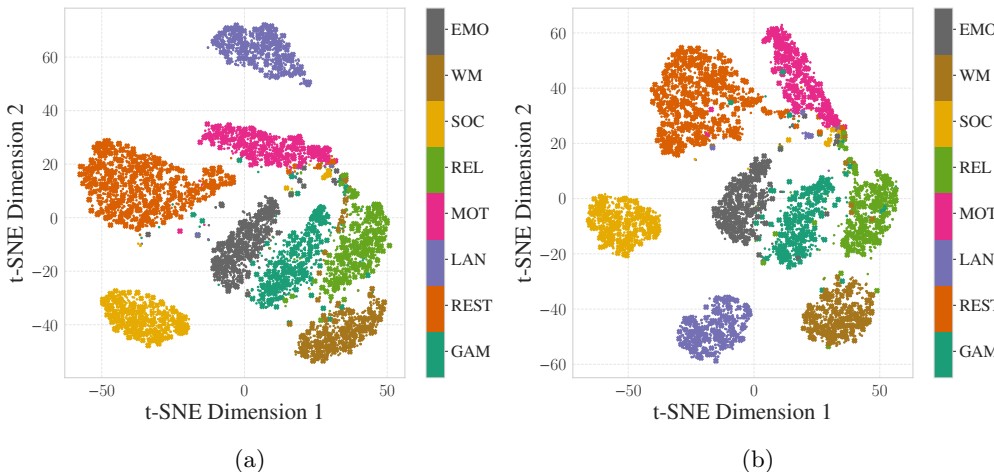

Figure 14: t-SNE of the embeddings created using (a) ADM+ and (b) ADM for $M = 0.3N$. X marks denote full-view points and dots partial-view points.

computational complexity analysis in Table 1. The interpolation based methods, NCCA and Dov et al. (2017), are the fastest as they only decompose small $M \times M$ matrices. The runtimes presented here do not include the distance computation between the features as we only do that once and reuse it for all methods.

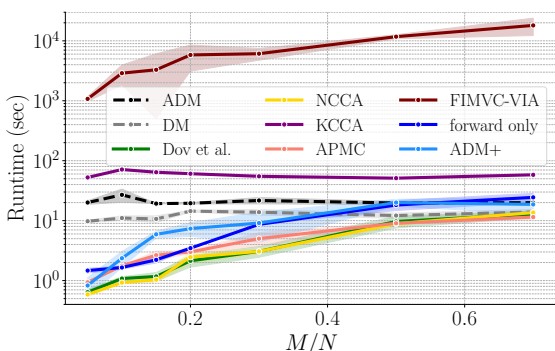

Figure 15: fMRI task runtime of different methods. The y-axis is presented in logarithmic scale.

## F  Competing Methods

Here we briefly present the competing methods used in the experimental part of the paper.

### F.1  Nyström Interpolation

Dov et al. (2017) suggested to use Nyström method to extend the ADM eigenvectors to fully viewed points. Let $\tilde{\psi}_i \in \mathbb{R}^M$ be a right eigenvector of $\widetilde{\mathbf{P}}_1 \widetilde{\mathbf{P}}_2$ with eigenvalue $\tilde{\lambda}_i$. Observe the following:

$$\widetilde{\mathbf{P}}_1 \widetilde{\mathbf{P}}_2 \tilde{\psi}_i = \tilde{\lambda}_i \tilde{\psi}_i \Rightarrow \tilde{\psi}_i = \frac{1}{\tilde{\lambda}_i} \widetilde{\mathbf{P}}_1 \widetilde{\mathbf{P}}_2 \tilde{\psi}_i \tag{77}$$

This relation gives rise to an interpolation scheme for the $k > M$ entry of the full eigenvector $\psi_i \in \mathbb{R}^N$, by using the extended kernel $\mathbf{P}_1^+$.

$$\psi_i(k) = \frac{1}{\widetilde{\lambda}_i} \sum_{m=1}^{M} [\mathbf{P}_1^+ \widetilde{\mathbf{P}}_2]_{k,m} \tilde{\psi}_i(m) \tag{78}$$

Then the eigenvectors and eigenvalues can be used to add the partially view samples to the embedding.

### F.2 KCCA

Kernel Canonical Correlation Analysis (KCCA) (Fukumizu et al., 2007) is an extension of linear CCA (Hotelling, 1936). Where we look for correlated projections of functions in two reproducing kernel Hilbert spaces (RKHSs). Rather than correlated directions in the feature space in CCA. By solving a generalized eigenvalue problem we get spanning coefficients in the RKHS for each function.

$$\begin{pmatrix} 0 & \mathbf{K}_1 \mathbf{K}_2 \\ \mathbf{K}_2 \mathbf{K}_1 & 0 \end{pmatrix} \begin{pmatrix} \alpha \\ \beta \end{pmatrix} = \lambda \begin{pmatrix} \mathbf{K}_1^2 & 0 \\ 0 & \mathbf{K}_2^2 \end{pmatrix} \begin{pmatrix} \alpha \\ \beta \end{pmatrix} \tag{79}$$

Where $\alpha, \beta$ are the projection coefficients. The original KCCA algorithm requires available samples from all views. Trivedi et al. (2010) proposed an extension of KCCA that allows one partial view. They use the graph Laplacian of the first view to smoothly impute the second view kernel. There is a closed-form solution for this imputation provided in their paper. Once the kernel is imputed KCCA can be applied to both kernels.

### F.3 NCCA

Nonparametric Canonical Correlation Analysis (NCCA) is a non-linear extension of CCA. This method estimates the probability density from and using kernel density estimation (KDE) and finds correlated projections in the function spaces of the different views. Michaeli et al. (2016) present the algorithm and show that the problem reduces to an eigensystem that can be solved using SVD. The SVD solution provides embedding for fully-viewed points which can be extended to partially-viewed points simply by using Nyström interpolation.

### F.4 Anchor Partial Multiview Clustering

Anchor Partial Multiview Clustering (APMC) (Guo & Ye, 2019) is a multiview clustering algorithm with partial view data. The algorithm first computes a low-dimensional embedding of the observations and them applied a clustering algorithm to it. For our sake, the embedding algorithm is of interest. As in ADM+, they use fully viewed samples as anchor points, but they do not focus on common information, and hence it is less clear what properties the embedding captures. To fuse the views, they compute $\mathbf{Q}_1^+$ and $\widetilde{\boldsymbol{Q}}_2$ and compute the fused affinity between all points and the anchors, $\mathbf{Q}^{\mathrm{APMC}} \in \mathbb{R}^{N \times M}$, by:

$$(\mathbf{Q}^{\mathrm{APMC}})_{i,j} = \begin{cases} (\mathbf{Q}_1^+)_{i,j}, & \text{if } i \notin \Omega_f \\ \frac{1}{2} \left[ (\mathbf{Q}_1^+)_{i,j} + (\widetilde{\boldsymbol{Q}}_2)_{i,j} \right], & \text{if } i \in \Omega_f \end{cases} , i = 1, ..., N, j = 1, ..., M \tag{80}$$

The embedding is then computed from the Laplacian Eigenmaps (Belkin & Niyogi, 2003) of the anchor graph (Liu et al., 2010) constructed from $\mathbf{Q}^{\mathrm{APMC}}$.

