# OpenReview forum: "Extracting Common Components from Partially Observed Views Using Diffusion Geometry"
_TMLR — Accepted by TMLR_

### Review · Reviewer_Dc4n · 2026-03-28

**Summary Of Contributions:**

This paper addresses the problem of extracting common components from multiview data under partial alignment, a practically relevant setting where some samples have missing measurements from one or more views. Existing methods either rely exclusively on fully aligned samples or require heuristic imputation/interpolation of missing data. The proposed method, ADM+, avoids such imputation and is able to utilize all available data, including partially observed samples, by using fully aligned samples as anchor points within a diffusion-geometric framework. The authors evaluate ADM+ across multiple experimental settings and demonstrate consistent advantages over existing kernel- and manifold-based baselines. In particular, the method shows greater robustness when the sampling distribution of the fully aligned set differs from that of the overall data. This robustness is also supported theoretically: the authors show that, in the infinite-sample limit, the proposed diffusion operator converges to an anisotropic diffusion process that emphasizes common components, with a density-ratio weighting mechanism that concentrates diffusion toward regions of high full-view density, thereby mitigating distributional discrepancies.

**Audience:**

Yes

**Audience Explanation:**

The paper addresses a clearly motivated and well-defined problem — common-component extraction from partially aligned multiview data — that is relevant to researchers working on manifold learning, multimodal data fusion, and missing data problems. The proposed approach combines diffusion geometry with an anchor-based strategy, which should be of interest to both the theoretical and applied communities within TMLR's readership. The paper also cites and builds upon recent works, indicating that this is an actively developing area.

**Claims And Evidence:**

Yes

**Claims Explanation:**

Yes, with minor reservations. The theoretical analysis (Theorem 1) provides a clear asymptotic justification for the proposed diffusion operator, showing convergence to an anisotropic diffusion process that emphasizes common components. The experimental evaluation covers three diverse domains (dynamical systems, synthetic images, and fMRI) with comparisons against multiple baselines, consistently supporting the claimed advantages. As noted in my major comment, some foundational assumptions — particularly the choice of the Euclidean metric and the Gaussian kernel on unknown manifolds — would benefit from more explicit justification, but the overall chain of evidence is convincing. The presentation could also be improved in terms of notation and accessibility (see minor comments), though these do not undermine the validity of the claims.

**Requested Changes:**

## Comment 1 (Major): Justification of the Euclidean metric and the role of the local approximation property
In Section 4.1, the kernel function (Eq. 2) is defined using a "chosen metric on the manifold," and the authors note that usual choices include the Euclidean distance or the geodesic distance "if it is known." However, given the problem formulation in Section 3 — where the observation functions f and g are explicitly assumed to be unknown — the manifold geometry is generally unknown as well. The paper would benefit from a clearer justification of why the Euclidean distance is a suitable choice in this setting. The authors also state that a key property of DM is that the diffusion distance "locally approximates the geodesic distance on the continuous manifold," but the practical implications of this fact are not fully elaborated. The following logical chain appears to underlie the methodology but is left largely implicit: (1) although the manifold is unknown, data points are isometrically embedded in Euclidean space; (2) locally, the Euclidean distance approximates the geodesic distance; (3) therefore, the Gaussian kernel with Euclidean distance yields a diffusion operator that asymptotically captures the intrinsic geometry; and (4) the resulting diffusion distance reflects the manifold structure. Making this reasoning explicit would strengthen the paper's exposition and help the reader understand how the local approximation property connects to the choice of metric and kernel.
## Comment 2 (Minor): Connection to heat kernels
The paper restricts attention to the Gaussian kernel "to simplify the discussion." Given that the theoretical framework is rooted in diffusion geometry, a brief discussion of the connection to heat kernels — which are the intrinsic objects on Riemannian manifolds and of which the Gaussian kernel serves as a discrete approximation in the epsilon -> 0 limit — would better situate the contribution within the broader literature and clarify the generalizability of the theoretical results.
## Comment 3 (Minor): Notation clarity
The notation could be improved in several respects. First, the paper does not visually distinguish between scalar and vector quantities (e.g., the hidden variables x, y, z and the observations s_i^{(l)} are all set in the same typeface), which can cause ambiguity given the complexity of the notation introduced later. Second, some variables are used before being explicitly defined — for instance, l appears in R^{N x l} (Section 3) as the embedding dimension but is not formally introduced at that point. Third, the use of both l (embedding dimension) and L (number of partial-view samples) is potentially confusing and could benefit from a different choice of symbol for one of them.
## Comment 4 (Minor): Accessibility of the presentation
Section 3 introduces the problem formulation directly in terms of manifolds (M_X, M_Y, M_Z) without a brief motivating explanation, while the relevant geometric background only appears in Section 4.1. A short informal description of the manifold assumption and its role in the modeling — even one or two sentences at the beginning of Section 3 — would make the transition smoother for readers less familiar with manifold learning. Similarly, while kernel functions are standard in the machine learning literature, a brief remark clarifying the role that kernels play in this work (measuring local similarity, defining transition probabilities, approximating the manifold geometry) would help readers quickly orient themselves within the framework.

---

> ### Author Response · Authors · 2026-05-05
> **Official Comment by Authors**
>
> We thank the reviewer for the careful reading, encouraging assessment, and constructive suggestions. In the revision, we focused on clarifying the metric and kernel choices, improving notation, and making the problem formulation more accessible.
>
> ### Responses to Requested Changes
>
> - **Justification of the Euclidean metric and local approximation.** Thank you for this helpful comment. We revised Section 4.1 to clarify the role of the metric used in the kernel construction. In the original text, the reference to the geodesic distance was intended to cover cases where a meaningful domain-specific metric is available, rather than to suggest that the exact geodesic distance on the unknown data manifold is known. For example, when the data points are SPD matrices, one may use the affine-invariant metric (Pennec et al., 2006) or log-Euclidean metric (Arsigny et al., 2006), as was done by Xiao et al. (2019). We revised the phrasing to make this point clearer.
>
>     In the absence of such prior geometric information, the Euclidean distance is typically used. This choice is standard in manifold learning: when data are sampled from a smooth manifold embedded in Euclidean space, graph operators constructed from Gaussian kernels based on Euclidean distances can be shown, with appropriate bandwidth and sampling, to converge to differential operators on the underlying manifold. In particular, Belkin and Niyogi (2003) showed that the graph Laplacian recovers the Laplace-Beltrami operator, while Coifman and Lafon (2006) showed that appropriately normalized diffusion operators recover the Laplace-Beltrami operator and the related Fokker-Planck operators.
>
>     We also revised the motivation for diffusion distance. Rather than emphasizing a direct connection to geodesic distance, which is more subtle, we now motivate the diffusion distance by its robustness to noise (Coifman and Lafon, 2006) and topological distortions (Bronstein et al., 2010), as well as its connection to random-walk connectivity. To further support its applicability, we cite notable recent works where diffusion distance and diffusion maps have been successfully used across different domains.
>
> - **Connection to heat kernels.** Thank you for the suggestion. We expanded the discussion in Section 4.1 to explain that the Gaussian kernel is further motivated by its approximation of the manifold heat kernel in the small-$\epsilon$ regime. This clarifies why the Gaussian kernel is natural in diffusion geometry and how the discrete graph diffusion relates to intrinsic diffusion operators on the manifold.
> - **Notation clarity.** We revised the notation to improve readability. Vectors are now consistently denoted in boldface, and the embedding dimension was renamed from $l$ to $r$ to avoid confusion with the number of partial-view samples $L$.
> - **Accessibility of the presentation.** We made the transition into the formal manifold model smoother. We expanded the discussion in Section 1 on manifold learning methods and the role of kernels in approximating the geometry of the data manifold. We added a concrete two-camera example in Section 1 and connected it to the latent-variable model in Section 3, explaining how $X$ represents shared scene content while $Y$ and $Z$ represent view-specific variability. We also added a forward-pointing paragraph at the beginning of Section 4 explaining the role of the diffusion-geometry preliminaries.

---

### Review · Reviewer_HkC4 · 2026-04-01

**Summary Of Contributions:**

**Summary**
The paper introduces ADM+, a manifold learning approach for recovering shared structure from partially observed multiview data. Its central idea is to treat the fully aligned samples as anchors, which allows the method to extract common information even when some views are missing, without relying on prior imputation or learning only from the fully aligned subset and then extending the embedding afterward. In addition to the method itself, the paper gives a diffusion-geometric perspective, shows that it is computationally more efficient than standard ADM, and evaluates it on examples from dynamical systems, synthetic rotating images, and fMRI-based task classification. Taken together, the paper addresses a relevant problem and offers a technically sound contribution.

**Strengths**
* S1: The paper addresses an important gap between fully aligned multiview methods and realistic partially aligned settings, and the motivation for common-component extraction rather than generic multiview fusion is well articulated.
* S2: ADM+ extends alternating diffusion to the partial-alignment setting through anchor-based coupling of the views, while still using both fully and partially observed samples instead of relying only on imputation or Nyström-style interpolation.
* S3: The diffusion interpretation and asymptotic analysis provide useful intuition for why the method should emphasize common structure and why it may be more robust to distribution mismatch than the forward-only alternative.
* S4: The experiments span synthetic and real data, and the image and fMRI results suggest ADM+ is often the strongest partial-view method, especially under distribution mismatch and limited alignment.

**Weaknesses**
* W1: The framework relies on assumptions such as conditional independence of view-specific factors and smooth manifold structure, but the paper gives limited discussion of failure cases when these assumptions are violated.
* W2: In the fMRI experiment, NCCA appears fairly competitive, so the practical advantage of ADM+ over the strongest baseline is not always as decisive as the narrative sometimes suggests.
* W3: The paper itself notes that the dependence on the number of aligned anchors $M$ is not theoretically understood, and this also weakens the practical guidance for using the method.
* W4: The presentation could be improved in several places. There are several minor writing, grammar, and notation issues, and some explanations could be made more precise, which at times makes an already technical paper harder to follow than necessary.

**Audience:**

Yes

**Audience Explanation:**

The paper studies a relevant problem in multiview learning that is likely to matter to readers working on representation learning, manifold learning, multimodal data analysis, and learning with missing or partially aligned views. Its combination of a new method, theoretical analysis, and experiments across different application settings gives it interest beyond a very narrow niche. Even for readers who may not use the method directly, the anchor-based extension of diffusion methods to partial multiview data is a useful idea and could inform related work in the area.

**Claims And Evidence:**

Yes

**Claims Explanation:**

The work reasonably supports its main claims through both analysis and experiments. Beyond presenting the algorithm, it also gives theoretical support through the diffusion-based interpretation and asymptotic results, which help explain why the method should recover shared structure and handle partial alignment effectively. The experimental section also covers several quite different settings, including synthetic and real data, and the comparisons with relevant baselines make the empirical case fairly convincing. There is still room for stronger ablations and a more detailed sensitivity analysis, but overall, the evidence is clear and sufficient to support the paper's central claims.

**Requested Changes:**

1. Add stronger ablations and sensitivity analyses, especially on the number of aligned samples $M$, degree/pattern of missingness, kernel scales, diffusion time, and embedding dimension.
2. Provide a more balanced discussion of the empirical results relative to the strongest baselines, especially NCCA, particularly in settings where the performance differences are modest.
3. Expand the discussion of assumptions and limitations, including cases with stronger view mismatch, weaker common structure, or violations of the conditional independence assumption.
4. Improve the presentation and reproducibility by addressing minor writing and notation issues, including a few more implementation and hyperparameter details in the main paper, and presenting runtime or memory comparisons more explicitly to better support the claimed efficiency benefits.

---

> ### Author Response · Authors · 2026-05-05
> **Official Comment by Authors**
>
> We thank the reviewer for their encouraging assessment of our work and their constructive feedback, which helped us improve the clarity, empirical support, and practical framing of the paper.
>
> ### Responses to Weaknesses
>
> - **W1: Conditional independence assumptions.** We now clearly mention and discuss the assumptions and limitations. We revised Section 3 to clarify that the conditional independence relation is part of the latent decomposition, and expanded the discussion in Section 8 regarding weak or absent common information (details below).
> - **W2: NCCA comparison.** We agree that NCCA is a strong baseline in the fMRI experiment, especially as $M/N$ increases. We revised Section 7.3 to present the comparison more carefully and to clarify the regimes where ADM+ has a clearer advantage (details below).
> - **W3: Dependency on $M$.** We agree that the dependence on the number and placement of aligned anchors $M$ is an important practical question. While a full theoretical characterization remains open, we added Figure 6b and accompanying discussion to provide additional empirical guidance (details below).
> - **W4: Presentation and notation.** We revised the notation, corrected several writing issues, added a forward-pointing paragraph at the beginning of Section 4, and added pointers to implementation, hyperparameter, and runtime details (details below).
>
> ### Responses to Requested Changes
>
> 1. **Ablations and sensitivity analysis.** Appendix E.2.2 includes sensitivity analyses for kernel scale, diffusion time, and embedding dimension. In the revision, we expanded the analysis of the number and pattern of aligned samples. Figure 6a studies random uniform and biased sampling patterns, Figure 6b compares grid-based and random uniform anchor selection for different values of $M$, and Figure 7 studies the effect of the aligned ratio $M/N$ in the fMRI experiment. These experiments provide empirical guidance on the role of $M$, missingness pattern, and key hyperparameters.
> 2. **Balanced discussion relative to NCCA.** We revised the empirical discussion in Section 7.3 to more carefully reflect the comparison with NCCA. We note that NCCA achieves comparable performance in the fMRI experiment and remains a strong baseline. At the same time, we clarify that ADM+ has clearer advantages in settings with fewer aligned samples and under distribution mismatch, where interpolation-based methods are more limited by their reliance on the fully aligned samples. This is consistent with the fact that interpolation-based extensions recover the embedding geometry only from the fully aligned subset, whereas ADM+ makes more efficient use of the available data by incorporating partial-view samples into the diffusion process.
> 3. **Assumptions and limitations.** Thank you for this important point. We expanded the discussion of assumptions and limitations in Sections 6.1 and 8. Section 3 now clarifies that the conditional independence relation is part of the model definition: $X$ is introduced as the latent variable capturing the information shared across the views, while $Y$ and $Z$ capture view-specific variability. Thus, conditional independence is built into the modeling decomposition rather than imposed as an additional empirical claim. Under this decomposition, dependence shared between the two views should be represented through $X$, rather than through residual dependence between $Y$ and $Z$. Therefore, the relevant limitation is not residual dependence between $Y$ and $Z$, but rather the regime in which the common component $X$ is weak relative to the view-specific variability, or effectively absent. In that case, extracting a meaningful common embedding becomes inherently difficult. We now discuss this limitation in Section 8 and note that Table 2 in Appendix E.1 includes empirical results related to weak common structure, while a corresponding theoretical treatment for the partial-view setting remains open. We also cite related works for standard ADM in the fully aligned setting. Finally, we added discussion in Section 6.1 on the diffeomorphism and smoothness assumptions used in Theorem 1, explaining that they should be viewed as idealized manifold-level assumptions that may hold only approximately in real data.
> 4. **Presentation and reproducibility.** We revised the notation and presentation throughout the paper. Vectors are now consistently denoted in boldface, and the embedding dimension was renamed from $l$ to $r$ to avoid confusion with the number of partial-view samples $L$. We also added a forward-pointing paragraph at the beginning of Section 4 to improve readability. To support reproducibility and efficiency claims, we added pointers to implementation and hyperparameter details and included a runtime comparison in Appendix E.2.3, with results in Figure 15.

---

### Review · Reviewer_9D7A · 2026-04-24

**Summary Of Contributions:**

This paper proposes ADM+, an extension of Alternating Diffusion Maps (ADM) to the *partially aligned* multiview setting, where only a subset of samples have measurements from all views. The key idea is to use fully aligned samples as anchor points within a diffusion geometry framework, constructing rectangular affinity matrices that relate the full-view and partial-view sets. The method avoids imputation of missing data or interpolation in the embedding space, a notable advantage over prior approaches such as Nyström-based ADM extensions (Dov et al., 2017) and NCCA (Michaeli et al., 2016).

The paper makes four main contributions:
1. the ADM+ algorithm itself, which naturally extends ADM-SVD to partial alignment;
2. a computationally efficient implementation reducing complexity from O(N^3) to O(NM^2), where M is the number of aligned samples;
3. a theoretical analysis (Theorem 1) showing that the continuous counterpart of ADM+ approximates an anisotropic diffusion process emphasising common components, with a density-ratio weighting that confers robustness to distributional discrepancies between aligned and unaligned samples; and
4. empirical evaluation on three tasks (coherent sets in dynamical systems, rotation angle estimation in synthetic images, fMRI task classification).

**Key strengths:**
- The method is clean and well-motivated from a diffusion geometry perspective. Using anchors to bridge full-view and partial-view sets is a natural and principled construction.
- The theoretical analysis is a genuine contribution: the asymptotic expansion (Theorem 1, equation 19) reveals a density-ratio weighting mechanism that provides an elegant explanation for the method's robustness to distributional mismatch.
- The comparison between ADM+ and the forward-only variant (Theorem 2 vs. Theorem 1) is insightful. It concretely shows the theoretical advantage of incorporating both forward and backward diffusion operators.
- The computational complexity improvement from O(N^3) to O(NM^2) is practically significant and well-explained (Appendix C).
- The experimental evaluation covers a reasonable range of domains, including a real-world fMRI application.

**Key weaknesses:**
- The paper is difficult to follow for readers not already immersed in the diffusion geometry literature. The formal machinery dominates, while concrete intuition and motivating examples are insufficient (detailed below).
- Several experimental design choices are not well justified, and some claims lack adequate statistical support.
- The paper's relationship to the broader multiview learning landscape (deep learning-based methods, contrastive approaches) is not discussed.

**Additional Comments:**

The paper is technically solid and addresses a genuine gap in the multiview manifold learning literature. The main barrier to acceptance is not the method or theory, but the *presentation*: the paper reads as if written primarily for experts in diffusion geometry, while the TMLR audience is broader. Addressing the critical items above , particularly the motivating example and improved narrative flow, would make the paper substantially more accessible without requiring new experiments or theory.

I would also note that the theoretical analysis, while rigorous, is an asymptotic result (as epsilon -> 0, N -> infinity). The paper could be more explicit about the gap between this continuous limit and the finite-sample, finite-bandwidth regime of the experiments. The simulations in Appendix D.3 partially bridge this gap, and it would be valuable to reference them more prominently from the main text.

**Audience:**

Yes

**Audience Explanation:**

The problem of extracting common components from partially aligned multiview data is practically relevant since full alignment is rarely available in real applications. The paper provides a theoretically grounded solution with a clean algorithmic formulation. Researchers working on multiview data analysis, manifold learning, and diffusion-based methods would find the contributions valuable.

That said, the paper's accessibility to the broader TMLR audience is limited by its presentation. The mathematical formalism, while precise, is not accompanied by sufficient intuition or concrete examples to make the key ideas accessible to readers outside the diffusion geometry community (see Requested Changes below).

**Broader Impact Concerns:**

None. The work is methodological and the applications (dynamical systems, synthetic images, fMRI) do not raise ethical concerns. No broader impact statement is required.

**Claims And Evidence:**

Yes

**Claims Explanation:**

**Leaning Yes**, with reservations on some claims.

The abstract makes the following claims:

1. *"Unlike existing methods, ADM+ does not require prior imputation of missing data or interpolation in the embedding space and makes use of all available data."*
**Supported.** The algorithm construction in Section 5 makes this clear: the rectangular kernel matrices directly incorporate all samples without an imputation step.

2. *"We provide a computationally efficient implementation, improving upon the O(N^3) time complexity of ADM."*
**Supported.** The complexity analysis in Appendix C is detailed and convincing, establishing O(NM^2 + NMq_1 + M^2q_2) time and O(NM) space.

3. *"A theoretical analysis showing that ADM+ approximates an anisotropic diffusion process that emphasizes common components."*

   **Supported.** Theorem 1 establishes the asymptotic expansion of the continuous operator S_epsilon, and the connection to the symmetric ADM operator of Shnitzer et al. (2019) is clearly articulated. The distributional robustness argument via the density ratio tilde{mu}^{(1)}/mu^{(1)} (Section 6.3) is compelling, and the simulations in Appendix D.3 provide useful visual confirmation.

4. *"Empirical evaluations [...] demonstrate that ADM+ achieves favorable performance compared to kernel- and manifold-based baselines."*
   **Partially supported.** ADM+ performs well across all three experiments, but:
   - In the coherent sets experiment (Section 7.1), results are averaged over only 10 repetitions, and no error bars or statistical tests are provided for the confusion matrices. The claim that "ADM+ indeed achieves coherent sets that are the most similar to ADM" is qualitative and based on visual inspection of Figure 3. The quantitative metrics (Silhouette score and dynamic isoperimetry) are deferred to Appendix E.1, where ADM+ achieves 0.409 +/- 0.016 vs. ADM's 0.380 +/- 0.034 in Silhouette score. This is a plausible but not statistically decisive difference given the overlapping standard deviations.
   - In the fMRI experiment (Section 7.3), ADM+ consistently outperforms other partial-view methods, which is convincing. However, all methods plateau near ~90% accuracy, and the practical significance of the differences at larger M/N is unclear.

5. *"ADM+ shows robustness to distributional discrepancies between aligned and unaligned samples."*
   **Supported.** The gamma-sweep experiment (Figure 6) is a well-designed test of this claim, and ADM+ degrades least as gamma increases.

**Requested Changes:**

## Critical

1. **Add a concrete running example early in the paper.** The paper defines "aligned" and "partially aligned" data in formal terms (Section 3) but never grounds these concepts in a specific, easy-to-understand example. The abstract mentions that "some samples have missing measurements from one or more views, and only a subset of the samples are fully aligned," but what does this mean concretely? For instance: "In a medical imaging study, each patient may undergo both an MRI scan and a CT scan (fully aligned), but some patients only have an MRI (partially aligned). The common component might be the underlying anatomy, while view-specific nuisance includes scanner-specific artefacts." A brief paragraph like this in the Introduction or at the start of Section 3 would substantially improve readability.

2. **Improve the narrative structure of Sections 4--5.** The paper currently moves through many definitions and prior methods (Section 4) before arriving at the proposed method (Section 5), but the reader lacks a clear sense of *why* each definition is needed until much later. Section 4 covers many definitions and describes previous approaches, but this is hard to parse without the big picture in place. Consider adding a brief forward-pointing paragraph at the start of Section 4 (e.g., "We will need the following ingredients: single-view diffusion operators to define affinities, and the ADM construction to see how alternating between views extracts common information. ADM+ will then extend this to the case where not all samples are observed in both views.").

3. **Strengthen statistical reporting in the experiments.** Specifically:
   - For the coherent sets experiment (Section 7.1), report the quantitative metrics (Silhouette score, dynamic isoperimetry) in the main text, not only in the appendix. These are the primary evidence for the claim.
   - For all experiments, clarify whether reported numbers are means, and over how many independent runs.
   - In Figure 6, the confidence intervals appear to be standard errors. Clarify whether these are SE or SD, and state the number of realizations (currently described as "generate 0 realizations" which appears to be a typo).

4. **Discuss the relationship to deep multiview learning methods.** The paper positions itself exclusively against kernel and manifold learning baselines. While this is the natural comparison class, the paper should at least acknowledge the existence of deep learning-based approaches to multiview learning with missing data (e.g., variational autoencoders for incomplete multiview data, contrastive multiview methods) and briefly discuss why kernel/manifold methods remain relevant despite the dominance of deep learning in many related areas. Without this, the positioning feels incomplete.

## Optional

5. **Clarify the role and sensitivity of M.** The method relies on M fully aligned anchor samples. The paper acknowledges (Section 8) that "understanding the method's dependency on M" is an open question, but given that this is arguably the most important practical parameter, more discussion is warranted. Figure 7 shows the effect of M/N on the fMRI task, but how sensitive is ADM+ to the *absolute* number of anchors M? Is there a minimum M below which the method degrades substantially? Even a brief empirical investigation on one of the existing experiments would be informative.

6. **Discuss the choice of kernel scale.** The kernel scale epsilon is set differently in each experiment (epsilon = 1 for coherent sets, validated via MAE for images, grid-searched for fMRI). The sensitivity analysis in Figure 12(b) is helpful but is confined to the appendix. A brief discussion of how to choose epsilon in practice would benefit practitioners.

7. **Clarify the assumptions for the theoretical analysis.** Theorem 1 requires that the manifolds are diffeomorphic (phi : M^{(1)} -> M^{(2)}) and that the densities are "sufficiently smooth." These are standard assumptions in the diffusion geometry literature, but the paper should discuss whether they are reasonable in the experimental settings. For instance, in the fMRI application, the two FCNs may not lie on diffeomorphic manifolds.

8. **Figure 1 could be more informative.** The illustration of the diffusion process (Figure 1) is helpful conceptually, but the colour coding and arrow conventions require careful reading of the caption. Consider adding axis labels or annotations directly on the figure.

9. **Minor typos and errors:**
   - "equation equation (6)" — duplicate word
   - "can also captures" — should be "can also capture"
   - "generate 0 realizations" (Section 7.2 / Figure 6 description) — appears to be a typo

---

> ### Author Response · Authors · 2026-05-05
> **Official Comment by Authors**
>
> We thank the reviewer for their detailed and constructive feedback that has helped us improve the clarity and strength of our manuscript.
>
> ### Responses to key weaknesses
>
> - **Running Example.** We added a concrete two-camera example (based on Section 7.2) to Sections 2 and 3, which clarifies the notions of alignment and partial alignment.
> - **Improved Statistical Reporting.** We strengthened the experimental reporting as requested (see details below), improving clarity and statistical support.
> - **Deep Learning Approaches.** We added a paragraph in Section 2 discussing deep learning approaches and positioning kernel/manifold methods within the broader scope.
>
> ### Response to the comment noting one partially supported claim
>
> > “Empirical evaluations [...] demonstrate that ADM+ achieves favorable performance compared to kernel- and manifold-based baselines.”
> >
>
> For the coherent sets experiment (Section 7.1), we moved the quantitative metrics from the appendix to the main text and added standard deviations to the confusion matrix. We emphasize that ADM serves as an oracle, as it uses fully aligned data; the goal of ADM+ is therefore not to surpass ADM, but to achieve comparable performance under partial alignment, which is supported by the results.
>
> Regarding the fMRI experiment, we clarified the interpretation of the plateau in Figure 7. In particular, ADM+ achieves performance close to ADM even at smaller $M/N$, demonstrating its ability to handle missing measurements. We also expanded the discussion comparing ADM+ and NCCA (Section 7.3), addressing cases where performance differences are small.
>
> ### Responses to the requested critical changes
>
> 1. **Running Example.** We added a concrete running example in the Introduction and Section 3, as suggested, to improve accessibility and intuition.
> 2. **Sections 4–5 Narrative.** We revised the beginning of Section 4 to include a forward-pointing paragraph explaining the role of the definitions and how they lead to ADM and ADM+.
> 3. **Statistical reporting.**
>     - Quantitative metrics for the coherent sets experiment are now included in the main text (Section 7.1).
>     - All reported values are now explicitly identified as means.
>     - Figure 6 now clearly states that the intervals are standard errors, and the number of repetitions (10) is specified, correcting the previous typo.
> 4. **Deep Learning Approaches.** We added a paragraph in Section 2 discussing deep learning approaches for incomplete multiview data and clarifying the relevance of kernel and manifold methods, particularly in low-data or resource-constrained settings.
>
> ### Responses to the requested optional changes
>
> 1. **Dependency on $M$.** We added results (Figure 6b) and discussion on the dependence on $M$. The results show that ADM+ performs well even for relatively small $M$, provided the aligned samples adequately cover the manifold. Performance depends on the missingness pattern, but ADM+ consistently outperforms competing methods, especially in more challenging settings.
> 2. **Scale Choice.** Regarding the kernel scale $\epsilon$, we follow standard practice. In the coherent sets experiment, $\epsilon=1$ was chosen based on the scale and uniformity of the data. In the other experiments, $\epsilon$ was selected via cross-validation using task-specific metrics. We also include a sensitivity analysis in Figure 12(b), which demonstrates that the method is robust to the choice of $\epsilon$ over a reasonable range.
> 3. **Theoretical Assumptions.** Regarding the assumptions in Theorem 1, we clarified their role in Section 6.1. In our experiments, there is an explicit one-to-one correspondence between samples in the two views, which can be viewed as a discrete analogue of the diffeomorphism assumption. The diffeomorphism thus serves as a continuous idealization of this alignment at the manifold level.
>
>     In practical settings such as the fMRI experiment, the transformations between views (e.g., different preprocessing pipelines) may not be globally smooth. However, they preserve the shared latent structure while introducing potentially non-smooth distortions in the view-specific components. Consequently, the mapping is expected to be smoother along the common component than along the view-specific variability. Since diffusion methods emphasize smooth structures, they can still recover the shared information even when the diffeomorphism assumption is only approximately satisfied. As a result, ADM+ remains effective even when smoothness does not hold for the view-specific components.
>
> 4. **Figure 1 Adjustment.** We adjusted Figure 1 to improve clarity by better separating the diffusion steps. We also explored adding additional annotations, but this introduced visual clutter and reduced readability.
>
> ### Minor typos and errors
>
> We thank the reviewer for identifying these typos; they have been corrected in the revision.

---

### Author Response · Authors · 2026-05-05
**General Author Response and Summary of Revisions**

We thank all reviewers for their careful reading and constructive feedback. We have uploaded a revised version of the manuscript, with changes marked in blue. The main revisions are summarized below.

- We added a concrete running example in Sections 1 and 3 to improve readability and to clarify the notions of full and partial alignment.
- We added a paragraph in Section 2 discussing deep learning methods for partially aligned multiview data, to better position our contribution within the broader literature.
- We added an opening paragraph to Section 4 to contextualize the diffusion-geometry preliminaries and explain how they relate to ADM+.
- We expanded the discussion in Section 6.1 on the practical interpretation of the diffeomorphism assumption used in the theoretical analysis.
- We moved the quantitative results (Silhouette score and dynamic isoperimetric ratio) for the coherent sets experiment from the appendix to the main text.
- We added Figure 6b and accompanying discussion to study the dependence on the number of aligned samples $M$ and the missingness pattern.
- We expanded the limitations discussion in Section 8, including the impact of the strength of the common information.

We believe these revisions improve the accessibility of the paper, strengthen the empirical reporting, and clarify the assumptions and practical scope of the proposed method.

---

### Decision · Action_Editor_6wHx · 2026-06-03

**Recommendation:** Accept as is

**Additional Comments:**

The reviewers are broadly positive, with final recommendations of Accept, Leaning Accept, and Accept. After considering the reviews, the author responses, and the revised manuscript, I agree with the reviewers' overall assessment that the paper is technically sound, relevant to the TMLR audience, and sufficiently supported by the presented theory and experiments.

I recommend acceptance as is. For the final version, I encourage the authors to further improve accessibility by adding intuitive explanation about diffusion maps and the associated geometric viewpoint, as this would help readers less familiar with diffusion geometry follow the work more easily. Please also ensure that the experimental section clearly reports the ADM+ hyperparameters used in each experiment.

**Audience:**

Yes

**Audience Explanation:**

Yes. All reviewers agreed that the paper would be of interest to TMLR audience, and I agree with this assessment.

The problem of extracting common components from partially aligned multiview data is relevant to researchers working on multiview learning, manifold learning, representation learning, multimodal data analysis, and missing/incomplete-view learning. The proposed ADM+ method should also be of interest to readers working on kernel methods, graph-based representation learning, diffusion geometry, and spectral methods.

**Claims And Evidence:**

Yes

**Claims Explanation:**

Yes. All three reviewers judged that the claims are supported with evidence. I agree with this assessment.

The submission proposes ADM+, an extension of Alternating Diffusion Maps to partially aligned multiview data, where only a subset of samples is observed in all views. The reviewers identified the anchor-based diffusion construction, the accompanying asymptotic analysis, and the empirical evaluation across multiple domains as sufficient support for the main claims.